# Combinations of maternal-specific repressive epigenetic marks in the endosperm control seed dormancy

**Hikaru Sato[1], Juan Santos-González[1], Claudia Köhler[1,2]\***

[1]Department of Plant Biology, Uppsala BioCenter, Swedish University of Agricultural Sciences and Linnean Centre for Plant Biology, Uppsala, Sweden; [2]Max Planck Institute of Molecular Plant Physiology, Potsdam-Golm, Germany

**Abstract** Polycomb Repressive Complex 2 (PRC2)-mediated trimethylation of histone H3 on lysine 27 (H3K27me3) and methylation of histone 3 on lysine 9 (H3K9me) are two repressive epigenetic modifications that are typically localized in distinct regions of the genome. For reasons unknown, however, they co-occur in some organisms and special tissue types. In this study, we show that maternal alleles marked by H3K27me3 in the *Arabidopsis* endosperm were targeted by the H3K27me3 demethylase REF6 and became activated during germination. In contrast, maternal alleles marked by H3K27me3, H3K9me2, and CHG methylation (CHGm) are likely to be protected from REF6 targeting and remained silenced. Our study unveils that combinations of different repressive epigenetic modifications time a key adaptive trait by modulating access of REF6.

## Introduction

Animals, fungi, and plants make use of two major epigenetic repressive modifications on histones that are destined to silence genes and transposable elements (TEs). The Polycomb repressive system applies the repressive trimethylation on lysine 27 of histone H3 (H3K27me3), which mainly serves to silence genes during defined stages of development (*Mozgova and Hennig, 2015*; *Schuettengruber et al., 2017*). In contrast, the repressive di- or trimethylation on lysine 9 of histone H3 (H3K9me2/3) is mainly associated with constitutively silenced heterochromatin, present in repeat-rich regions and TEs (*Saksouk et al., 2015*; *Wendte and Schmitz, 2018*). Repression mediated by the Polycomb system is transient and dynamically regulated, while repression mediated by H3K9me2/3 is generally stable during development (*Mozgova and Hennig, 2015*; *Saksouk et al., 2015*; *Schuettengruber et al., 2017*; *Wendte and Schmitz, 2018*). The establishment of H3K27me3 is mediated by the evolutionary conserved Polycomb repressive complex 2 (PRC2). In plants, there are distinct PRC2 complexes that are active during different stages of plant development (*Mozgova and Hennig, 2015*). In *Arabidopsis*, the EMBRYONIC FLOWER (EMF) and VERNALIZATION (VRN) PRC2 regulate sporophyte development, while the FERTILIZATION-INDEPENDENT SEED (FIS) PRC2 is active in the central cell of the female gametophyte and the descendent endosperm (*Mozgova and Hennig, 2015*). The establishment of H3K9me2 in *Arabidopsis* depends on the SU(VAR)3–9 homologous proteins (SUVHs), KRYPTONITE (or SUVH4), SUVH5 and SUVH6 that act in a feedback loop with CHG methylation (CHGm) established by the CHROMOMETHYLTRANSFERASE 3 (CMT3) (*Kenchanmane Raju et al., 2019*; *Zhang et al., 2018*). In animals and plants, H3K27me3 and H3K9me2 are generally exclusive repressive marks (*Wiles and Selker, 2017*); however, they co-occur in certain tissue types like the plant endosperm (*Klosinska et al., 2016*; *Moreno-Romero et al., 2016*; *Weinhofer et al., 2010*), as well as in specific organisms as filamentous fungi and bryophytes (*Carlier et al., 2020*; *Montgomery et al., 2020*). Furthermore, redistribution of H3K27me3 to heterochromatic regions occurs

*For correspondence: claudia.kohler@slu.se

Competing interest: The authors declare that no competing interests exist.

when constitutive heterochromatin is disrupted, indicating that specific features of heterochromatin prevent PRC2 recruitment (*Deleris et al., 2012*; *Jamieson et al., 2016*; *Peters et al., 2003*; *Wiles and Selker, 2017*).

Erasure of H3K27me3 and H3K9me2 is mediated by different types of JUMONJI lysine demethylases (KDMs) that belong to the KDM4 and KDM3 type, respectively (*Cheng et al., 2020*). RELATIVE OF EARLY FLOWERING 6 (REF6) and EARLY FLOWERING 6 (ELF6) function redundantly as major erasers of H3K27me3 during *Arabidopsis* development (*Lu et al., 2011*; *Yan et al., 2018*). The zinc-finger (ZnF) domains of REF6 recognize the sequence motif CTCTGYTY (Y = T or C) (*Cui et al., 2016*; *Li et al., 2016*). Binding of REF6 to this motif is negatively affected by non-CG methylation (*Qiu et al., 2019*), revealing cross-talk between different epigenetic modifications.

Seed development in flowering plants is initiated by a double fertilization event, where one of the sperm cells fertilizes the haploid egg, giving rise to the diploid embryo and the other sperm cell fertilizes the diploid central cell, initiating development of the triploid endosperm (*Zhou and Dresselhaus, 2019*). The endosperm regulates the transfer of nutrients to the developing embryo, similar as the placenta in mammals. (*Costa et al., 2012*; *Lee et al., 2010*; *Linkies et al., 2009*). The endosperm furthermore regulates seed dormancy, a process preventing germination of seeds under favourable conditions (*Nonogaki, 2019*). Both processes have been linked to genomic imprinting, an epigenetic phenomenon causing parent-of-origin specific gene expression (*Batista and Kohler, 2020*; *Gehring and Satyaki, 2017*; *Piskurewicz et al., 2016*). Imprinted genes are epigenetically modified during gametogenesis, establishing parental-specific expression patterns that are maintained after fertilization. Genomic imprinting leading to maternally biased expression is predominantly connected to DNA demethylation of maternal alleles mediated by the DNA glycosylase DEMETER (*Gehring et al., 2009*; *Hsieh et al., 2011*; *Park et al., 2016*). In contrast, paternally biased expression is predominantly connected to the presence of H3K27me3, H3K9me2, and CHGm on maternal alleles (*Klosinska et al., 2016*; *Moreno-Romero et al., 2019*; *Moreno-Romero et al., 2016*; *Moreno-Romero et al., 2017*). The FIS-PRC2 establishes H3K27me3 in the central cell, which is maintained on the maternal alleles after fertilization. Loss of FIS-PRC2 function impairs CHGm establishment, indicating that H3K27me3 is the primary imprinting mark and CHGm and H3K9me2 are recruited after H3K27me3 establishment (*Moreno-Romero et al., 2019*). The biological meaning of the co-occurance of H3K27me3, H3K9me2, and CHGm on the maternal alleles of paternally expressed genes (PEGs) is unclear. In this study, we reveal that genes marked by H3K27me3 on their maternal alleles are targeted by REF6 and become reactivated during germination. In contrast, REF6 fails to activate genes marked by H3K27me3, H3K9me2, and CHGm, consistent with the finding that DNA methylation prevents REF6 binding (*Qiu et al., 2019*). Among those genes with triple repressive marks was the *ABA INSENSITIVE 3* (*ABI3*) gene, a known positive regulator of seed dormancy (*Liu et al., 2013*; *Shu et al., 2016*; *Tian et al., 2020*). Genes activated by REF6 are enriched for ethylene signaling, a process activated during germination. Consistently, loss of *REF6* causes increased dormancy, as recently reported (*Chen et al., 2020*). Together, our study unveils a mechanism that employs the combination of different repressive epigenetic modifications to time a crucial developmental process.

## Results

### Genes marked by single H3K27me3 are expressed during germination

To understand the functional relevance of different combinations of repressive epigenetic modifications in the endosperm, we analyzed gene ontologies (GOs) of genes marked with single H3K27me3, single H3K9me2, and both modifications (referred to as H3K27me3/H3K9me2) using AgriGO v2.0 (http://systemsbiology.cau.edu.cn/agriGOv2/; *Tian et al., 2017*; *Figure 1—figure supplement 1*). Genes marked by single H3K9m2 were not enriched for any GO category and genes marked by H3K27me3/H3K9me2 only for one; in contrast, genes marked with single H3K27me3 were significantly enriched for many GOs (*Figure 1—figure supplement 1*). Interestingly, among those GOs was an enrichment of genes related to ethylene signaling (*Figure 1—figure supplement 1*; *Dubois et al., 2018*), a pathway which is activated in the endosperm tissue during germination to allow root protrusion through cell loosening (*Linkies and Leubner-Metzger, 2012*; *Linkies et al., 2009*) and known to release seed dormancy (*Beaudoin et al., 2000*; *Corbineau et al., 2014*). Supporting the GO enrichment, we found that genes with single H3K27me3 significantly overlapped with ethylene-inducible

genes identified in a previous study (*Das et al., 2016*), while genes with single H3K9me2 and H3K27me3/H3K9me2 did not (*Figure 1a*). We analyzed the expression of ethylene-inducible genes marked by single H3K27me3 during germination in the endosperm using previously published data (*Dekkers et al., 2013*). We found that ethylene-inducible genes with single H3K27me3 were induced during germination, especially around the time point of endosperm rupture (p<0.05, Wilcoxon test) (*Figure 1b*), while there were no significant expression differences between different time points for genes with single H3K9me2 and genes with both H3K27me3/H3K9me2. Increased gene expression in the endosperm during germination was not restricted to ethylene inducible genes marked by H3K27me3, but also detectable for all H3K27me3 marked genes (*Figure 1—figure supplement 2*). This data reveal that genes with single H3K27me3 are initially suppressed during early endosperm development and become reactivated during germination, while genes with single H3K9me2 and genes with H3K27me3/H3K9me2 remain continuously suppressed throughout endosperm development and germination. This implies the presence of a molecular mechanism that allows to differentially regulate the expression of the three groups of epigenetically marked genes during germination.

## Genes with double H3K27me3/H3K9me2 modifications are suppressed through CHG methylation on REF6-binding sites during germination

Recent work revealed that the presence of CHG methylation (referred to as CHGm) prevents binding of the H3K27me3 histone demethylase RELATIVE OF EARLY FLOWERING 6 (REF6/JMJ12) to its target motif CTCTGYTY (Y = T or C) (*Qiu et al., 2019*). We recently showed that genes with dual H3K27me3/H3K9me2 modifications are marked by CHGm in the endosperm and that CHGm is established in the central cell (*Moreno-Romero et al., 2019*). We thus hypothesized that the presence of CHGm may prevent REF6 from targeting and activating those genes. To test this hypothesis, we identified genes with REF6 binding sites in their gene bodies and associated the presence of the REF6 motif, CHGm status, and expression during germination. We found that REF6 binding sites within genes that were marked either by single H3K9me2 or by H3K27me3/H3K9me2 were more frequently CHG methylated in the central cell (*Figure 2a*) and endosperm (*Figure 2—figure supplement 1*) compared to single H3K27me3 marked genes (*Moreno-Romero et al., 2016*; *Park et al., 2016*). Importantly, genes marked by H3K27me3/H3K9me2 were frequently marked by CHGm at position five within the REF6-binding site (*Figure 2b*). Methylation at position five was found to be sufficient to abolish the affinity of REF6 for this target site, while methylation at position three was less effective (*Qiu et al., 2019*). Consistent with the hypothesis that CHGm on REF6 binding sites prevents REF6 targeting and thus gene activation, we found that genes with non-methylated REF6 binding sites were significantly more frequently activated during germination than genes with methylated REF6 binding sites (*Figure 2c*). Together, this data supports the idea that the methylation status of REF6-binding sites affects the reactivation of genes in the endosperm during germination. While H3K27me3 marked genes can be activated during germination, genes with H3K27me3, H3K9me2, and CHGm fail to be activated since REF6 (or homologous proteins) recruitment is blocked by DNA methylation.

## REF6 is required to promote germination

Loss of *REF6* was recently reported to cause increased dormancy (*Chen et al., 2020*). There are two REF6 orthologs in *Arabidopsis thaliana*, EARLY FLOWERING 6 (ELF6), and JUMONJI 13 (JMJ13). The DNA-binding domains of REF6 and ELF6 share amino acids that are required for binding to REF6-sequence motifs (*Qiu et al., 2019*) contrasting with lack of conservation of those sites in JMJ13 (*Figure 3—figure supplement 1*). This suggests that ELF6 acts redundantly with REF6 to promote germination. We tested two independent mutant alleles of *ref6* and *elf6* that all showed late germination phenotypes, but the *ref6* mutants were more strongly delayed than *elf6* (*Figure 3—figure supplement 2*). To test whether there are genetic redundancies between *REF6*, *ELF6*, and *JMJ13* in controlling germination, we analyzed double *elf6 ref6* and triple *elf6 ref6 jmj13* mutants (*Yan et al., 2018*) for their ability to germinate. The *elf6 ref6* double was slightly, but significantly, more delayed compared to the *ref6* single mutant, while the *elf6 ref6 jmj13* triple mutant behaved like the *elf6 ref6* double mutant (*Figure 3a*). This data reveal that REF6 is the major factor releasing seed dormancy and promoting germination and that ELF6 is partially redundant with REF6, consistent with the conserved DNA binding domain (*Figure 3—figure supplement 1*). Prolonged after-ripening or stratification of *ref6* seeds decreased primary dormancy and promoted germination (*Figure 3—figure supplement*

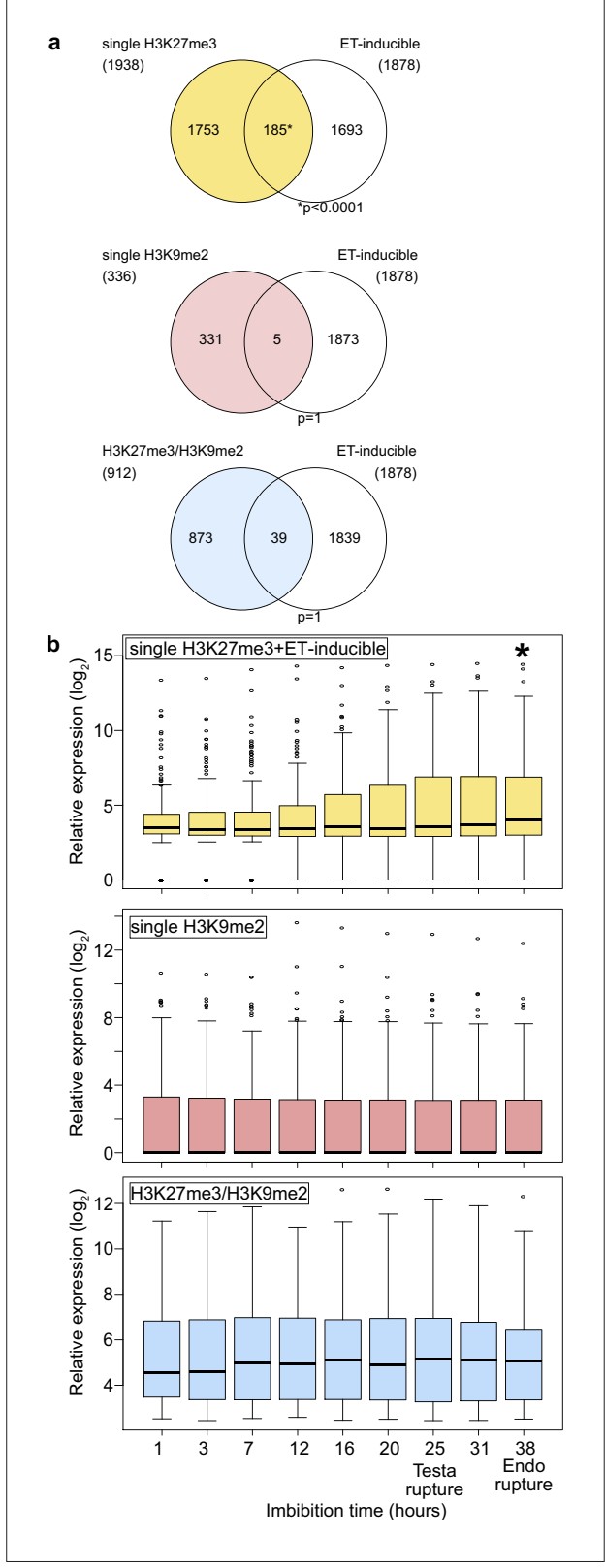

**Figure 1.** Genes marked by H3K27me3 become activated in the endosperm during germination. (**a**) Venn diagrams show overlaps between three groups of genes with different patterns of histone modifications and ethylene (ET)-inducible genes in seedlings. ET-inducible genes identified in a previous study (*Das et al., 2016*) were compared with genes marked by the indicated histone modifications in the developing endosperm (*Moreno-*

*Figure 1 continued on next page*

*Figure 1 continued*

***Romero et al., 2016***). Asterisks indicate significant overlap (p<0.0001, Fisher's exact test). (**b**) Box plots show relative expression profiles of genes marked by the indicated histone modifications in the endosperm during germination. Time-course expression profiles of ethylene inducible genes with single H3K27me3 (top; 185 genes), single H3K9me2 (middle; 336 genes), and H3K27me3, H3K9me2 and CHG methylation (bottom; 227 genes). Expression data are endosperm-specific transcriptome data during germination (***Dekkers et al., 2013***). The time points of testa rupture and endosperm (Endo) rupture are shown. Asterisks indicate significant differences compared to the time point at 1 hr of imbibition (p<0.01, Wilcoxon test).

The online version of this article includes the following figure supplement(s) for figure 1:

**Figure supplement 1.** Enrichment of GOs among genes with single H3K27me3 and double H3K27me3/H3K9me2 on the maternal genome in the endosperm.

**Figure supplement 2.** Expression profiles of all genes with single H3K27me3 during germination.

*3*), similar to other dormancy-related mutants (***MacGregor et al., 2019***; ***Piskurewicz et al., 2016***; ***Zheng et al., 2012***). Loss of maternal, but not paternal *REF6* caused a significant delay in germination (***Figure 3b***), revealing that REF6 acts in maternal tissues. Nevertheless, combined loss of maternal and paternal *REF6* had a stronger effect than loss of the maternal *REF6* (***Figure 3b***), indicating that REF6 also acts after fertilization. To test whether the maternal effect of *ref6* is caused by a functional role of REF6 in maternal sporophytic tissues or in the female gametophyte, we analyzed the phenotypes of F1 seeds generated by crossing maternal heterozygous *ref6-1* mutant plants with paternal wild-type plants. The F1 seeds showed significantly delayed germination compared to wild-type seeds, with germination frequencies being in between that of wild-type seeds and F1 seeds of homozygous *ref6* mutants crossed with wild-type pollen (***Figure 3c***). Since *REF6* is not imprinted (***Hsieh et al., 2011***; ***Pignatta et al., 2014***), this data strongly suggests that REF6 acts before fertilization in the female gametophyte.

## Endosperm-specific expression of *REF6* is important for germination

Our data suggests a major role of REF6 in the endosperm to promote the expression of genes required for releasing dormancy. To further challenge this model, we tested whether expression of *REF6* in the micropylar endosperm was sufficient to suppress the germination delay of *ref6* mutants. We expressed *REF6* in the *ref6* mutant under control of the *EARLY-PHYTOCHROME-RESPONSIVE1* (*EPR1*) promoter that is specifically expressed in the micropylar endosperm during germination (***Dubreucq et al., 2000***; *Figure 4—figure supplement 1*). Two independent transgenic *EPR1::REF6; ref6* lines showed a significant suppression of the *ref6* germination delay (***Figure 4a***), indicating that expression of *REF6* in the micropylar endosperm is functionally required to release dormancy and promote germination. We also expressed *REF6* under control of the *TREHALOSE-6-PHOSPHATE SYNTHASE 1* (*TPS1*) promoter that is specifically active in the embryo during germination (***Bae et al., 2009***; *Figure 4—figure supplement 1*). However, *TPS1::REF6* did not suppress the germination delay of *ref6* (***Figure 4b***), indicating that REF6 has no functional role in the embryo during germination. In support of this notion, dissected *ref6* embryos did not show any delay in growth compared to dissected wild-type embryos (***Figure 4c,d***). Together, this data supports our model that REF6 is required in the endosperm to promote the expression of genes required for releasing dormancy.

## REF6 activates genes marked by single H3K27me3 during germination

To identify the genes controlled by REF6 during germination, we profiled the transcriptome of *ref6* mutant endosperm during germination. We found more than twice as many significantly downregulated than upregulated genes in *ref6* compared to wild type (1471 downregulated [$Log_2FC \leqq -1$, p<0.05] vs 655 upregulated [$Log_2FC \geq 1$, p<0.05]; *Figure 5—source data 1*), consistent with the proposed function of REF6 to activate genes during germination. Genes with single H3K27me3 and unmethylated REF6-binding sites were significantly enriched among downregulated but not upregulated genes (***Figure 5a,b***), supporting the model that REF6 activates genes with single H3K27me3 through the unmethylated REF6-binding sites. Furthermore, there was no significant enrichment of genes with single H3K9me2 among deregulated genes in *ref6* (***Figure 5—figure supplement 1***). Additionally, while genes with double H3K27me3/H3K9me2 and CHG methylated REF6-binding sites were not significantly enriched among deregulated genes, genes with double H3K27me3/

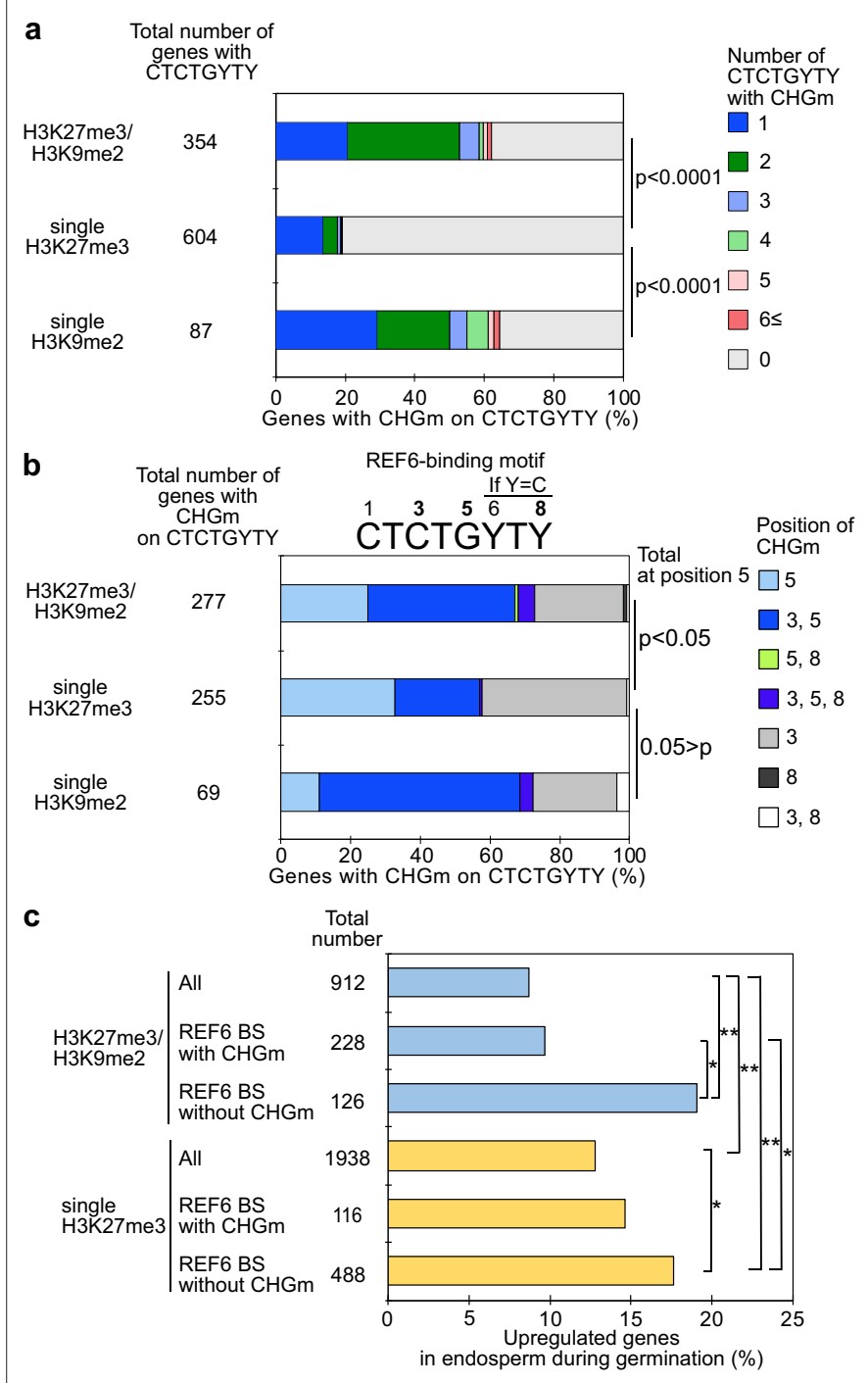

**Figure 2.** Gene activation during germination depends on CHG methylation (CHGm) on REF6 binding sites (REF6-BS). (**a**) Plot shows the percentage of REF6-BS with CHGm in the three groups of genes with different combinations of histone modifications. CHGm on REF6-BS in gene bodies was determined based on published DNA methylome data (***Park et al., 2016***). p values were calculated using pairwise Fisher's exact test and Benjamini–Hochberg correction. (**b**) Plot shows the percentage of CHGm on different cytosine positions of REF6-BS in the three groups of genes with different combinations of histone modifications. CHGm on REF6-BS in gene bodies was determined based on published DNA methylome data (***Park et al., 2016***). Numbers above the REF6-BS indicate possible positions of DNA methylation, of which positions 3, 5, and 8 can be targeted by CHGm. p values were calculated using pairwise Fisher's exact test and Benjamini–Hochberg correction and indicate significance for CHGm on position five in REF6-BS. (**c**) Plot shows the percentage of upregulated genes with double H3K27m3/H3K9me2 and

*Figure 2 continued on next page*

*Figure 2 continued*

single H3K27me3 in the endosperm during germination depending on CHGm on REF6-BS. Gene expression is based on previously published endosperm-specific transcriptome data during germination (*Dekkers et al., 2013*). Compared were expression levels between 1 and 38 hours of imbibition (1≦Log$_2$FC, p<0.05). Asterisks indicate significant differences between two categories (*p<0.05; **p<0.01; pairwise Fisher's exact test with Benjamini–Hochberg correction).

The online version of this article includes the following figure supplement(s) for figure 2:

**Figure supplement 1.** Percentage of REF6-binding sites (REF6-BS) with CHG methylation (CHGm) (based on published data [*Moreno-Romero et al., 2016*]) among three groups of genes with different combinations of histone modifications (based on published data [*Moreno-Romero et al., 2016*]) in developing endosperm.

H3K9me2 and unmethylated REF6-binding sites were slightly enriched among downregulated genes, although the enrichment was not significant (p=0.0568, pairwise Fisher's exact test) (*Figure 5—figure supplement 2*). The 59 genes with single H3K27me3 and REF6-binding sites that were downregulated in *ref6* (*Figure 5—figure supplement 3*) were significantly enriched for germination- and ethylene-inducible genes, but not for gibberellin-inducible genes according to previous transcriptome analyses (*Cao et al., 2006*; *Das et al., 2016*; *Dekkers et al., 2013*; *Figure 5c*). These findings are consistent with the idea that REF6 promotes germination by activating germination-inducible genes acting in the ethylene pathway. Among the candidate REF6 target genes (*Figure 5—figure supplement 3*) was *ETHYLENE RESPONSE DNA BINDING FACTOR 4* (*EDF4*) that was previously reported to mediate ethylene responses (*Alonso et al., 2003*; *Chen et al., 2015*). We confirmed significant suppression of *EDF4* in the *ref6* mutant endosperm (*Figure 5d*), supporting the proposed endosperm-specific function of REF6 during germination (*Figure 4a,b*). We furthermore found three genes encoding for ethylene biosynthesis enzymes converting the ethylene precursor 1-aminocyclo propane-1-carboxylic acid (ACC) to biologically active ethylene (ACC oxidases, ACOs) significantly downregulated in *ref6* (*Figure 5—figure supplement 3*). *ACO* genes were not marked by H3K27me3 in the 4 days after pollination (DAP) endosperm, suggesting that REF6 activates these genes at an earlier timepoint. REF6 was shown to target the ABA catabolism genes *CYP707A1* and *CYP707A3*, but not *CYP707A2* during seed development (*Chen et al., 2020*). We also found *CYP707A1* and *CYP707A3* slightly, but significantly suppressed in *ref6* endosperm (*Figure 5—figure supplement 3*), consistent with the presence of REF6-BS in *CYP707A1* and *CYP707A3*, but not *CYP707A2*. Like for *ACO* genes, we did not detect H3K27me3 on all three genes in 4 DAP endosperm (*Moreno-Romero et al., 2016*), suggesting that REF6 acts on these genes before this timepoint. To further test the contribution of the ethylene and ABA pathways to the delayed germination phenotypes in *ref6* mutants, we analyzed germination of *ref6* in the presence of the ethylene precursor 1-a minocyclopropane-1-carboxylic acid (ACC) and the ABA biosynthesis inhibitor fluridone, which both were reported to induce seed germination (*Linkies et al., 2009*; *Martinez-Andujar et al., 2011*). Treatment with only ACC or fluridone partially suppressed the *ref6* delayed germination phenotype, but the germination ratios were significantly lower than non-treated wild type after stratification (*Figure 5—figure supplement 4*). In contrast, treatment of *cyp707a2* mutants with fluridone resulted in comparable germination frequencies as for wild-type seeds without treatment (*Figure 5—figure supplement 4*), suggesting that induction of the ABA pathway can be effectively suppressed by fluridone, in line with previous data (*Martinez-Andujar et al., 2011*). The failure of fluridone to suppress the *ref6* mutant phenotype is also consistent with previous findings showing that a mutation in the ABA biosynthesis gene *ABA2* does not completely rescue the delayed germination phenotype of *ref6-1* after stratification (*Chen et al., 2020*). Importantly however, treatment with both ACC and fluridone could completely suppress the delayed germination of *ref6-1* after stratification (*Figure 5—figure supplement 4*). This data support the idea that loss of REF6 causes failure to activate the ethylene pathway (*Figure 5c*) and at the same time causes induction of the ABA pathway as previously shown (*Chen et al., 2020*).

Together, our data support a model where REF6 is required to activate genes with single H3K27me3 and unmethylated REF6-binding sites in the endosperm during germination, among them genes in the ethylene and ABA catabolism pathway.

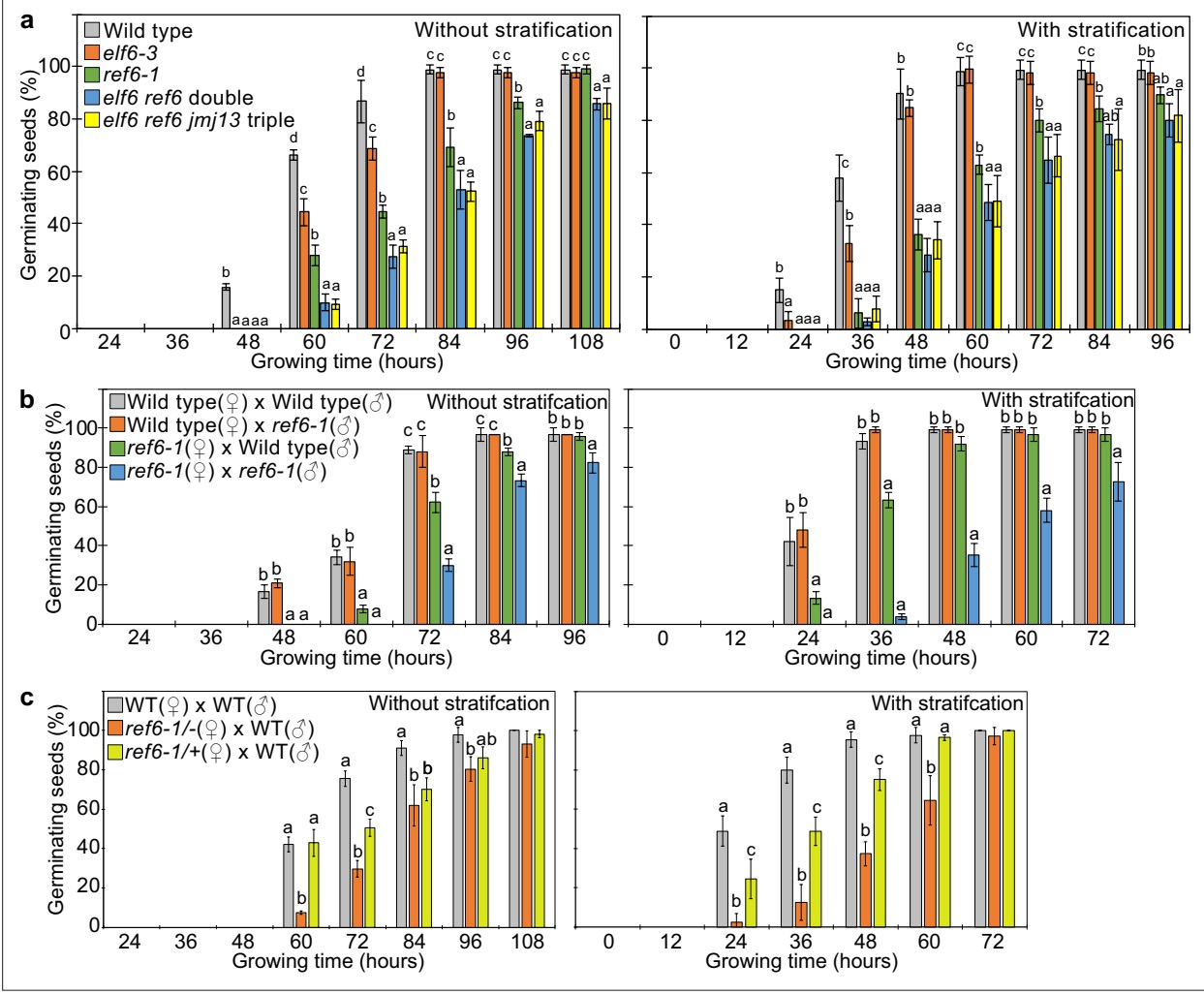

**Figure 3.** REF6 is the major H3K27me3 demethylase controlling seed dormancy. (**a**) Plots show the percentage of germinating seeds of wild type, *elf6-3* single, *ref6-1* single, *elf6-3 ref6^C* double, and *elf6-3 ref6^C jmj13^G* triple mutants with and without stratification. Error bars indicate SD from three biological replicates (n = 120 total). Letters above bars indicate significant differences among genotypes (p<0.05, Tukey's multiple range test). (**b**) Germination phenotypes of seeds generated by reciprocal crosses between wild type and *ref6-1*. Plots show the percentage of germinating seeds with and without stratification. Error bars indicate SD from three biological replicates (n = 120 total). Letters above bars indicate significant differences among genotypes (p<0.05, Tukey's multiple range test). (**c**) Plots show the percentage of germinating seeds with and without stratification of wild type (WT), F1 seeds generated by crossing maternal heterozygous or homozygous *ref6-1* and paternal wild type. Error bars indicate SD from three biological replicates (n = 90 total). Letters above bars indicate significant differences among genotypes (p<0.05, Tukey's multiple range test).

The online version of this article includes the following figure supplement(s) for figure 3:

**Figure supplement 1.** Alignment of the predicted DNA-binding motifs of REF6, ELF6, and JMJ13.

**Figure supplement 2.** Germination phenotypes of *ref6*, *elf6* single mutants.

**Figure supplement 3.** Seed dormant phenotypes of *ref6* mutants.

## SUVH4/5/6 and CMT3 are required to promote germination

Our data indicate that REF6 is able to activate genes marked by single H3K27me3, but fails to access genes marked by H3K27me3/H3K9me2/CHGm due to the block of REF6 targeting by CHGm. The H3K9me2 modification is established by the redundantly acting SUVH4/5/6 histone methyltransferases and recruits CMT3 to establish DNA methylation in CHG context (*Du et al., 2012*; *Ebbs and Bender, 2006*; *Jackson et al., 2002*). Since the *suvh456* mutant is strongly depleted for CHGm (*Stroud et al., 2014*), we speculated that in *suvh456* REF6 targets should become accessible and activated. Previous work revealed that the *suvh45* double mutant has increased dormancy (*Zheng et al., 2012*). Similarly, we found that *suvh456* required prolonged after-ripening time and had reduced germination rates

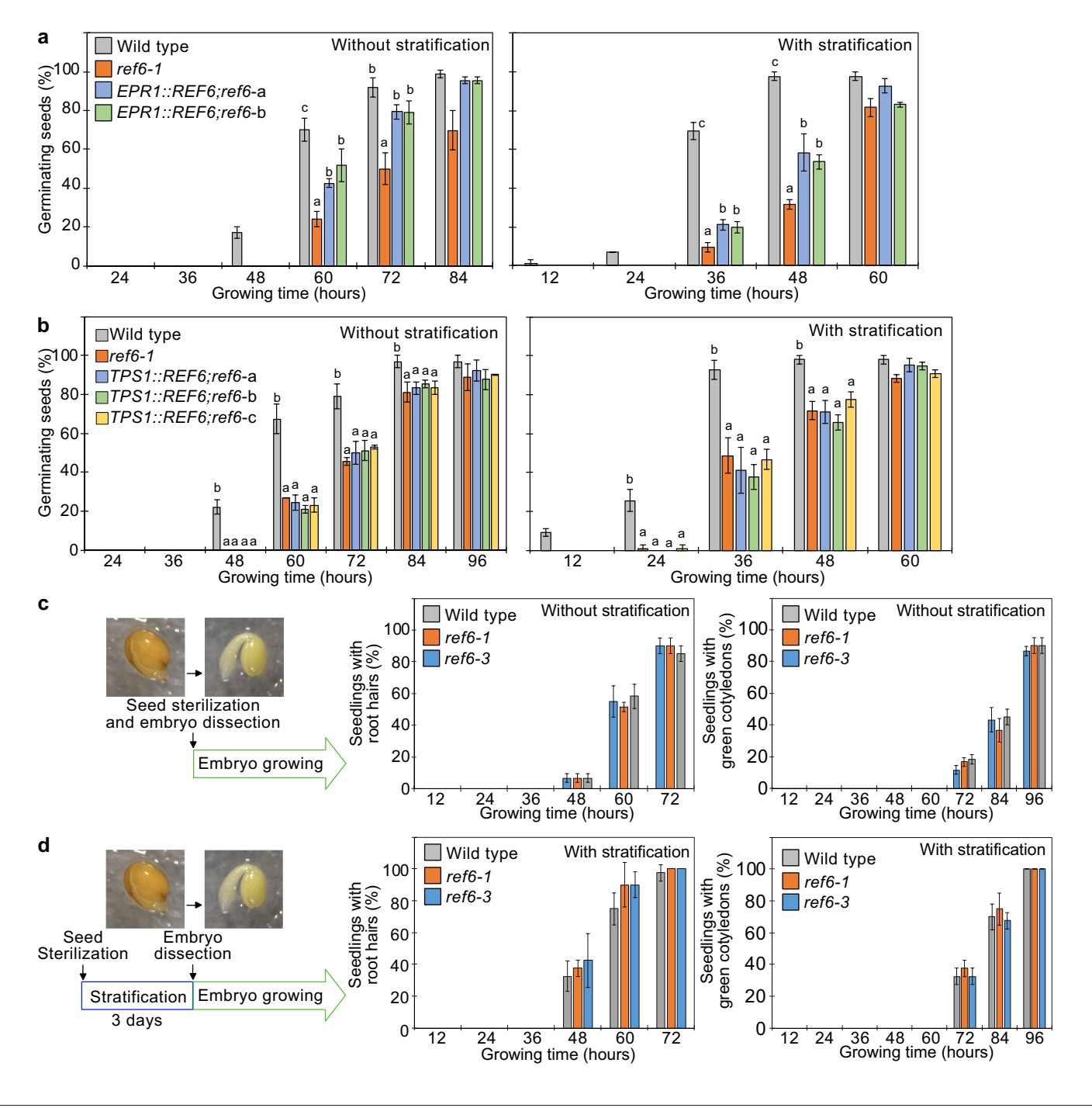

**Figure 4.** *REF6* expression in the endosperm promotes germination. (**a**) Germination phenotypes of the *ref6-1* mutant and transgenic lines in the *ref6-1* background expressing *REF6* under control of the micropylar endosperm-specific *EPR1* promoter. Plots show the percentage of germinating seeds of wild-type, *ref6-1,* and complemented lines (*EPR1::REF6;ref6-1*-a and b) with and without stratification. Error bars indicate SD from three biological replicates (n = 120 total). The letters above the bars indicate significant differences among plant lines (p<0.05, Tukey's multiple range test). (**b**) Germination phenotypes of the *ref6-1* mutant and transgenic lines in the *ref6-1* background expressing *REF6* under control of the embryo-specific *TPS1* promoter during germination. Plots show the percentage of germinating seeds of wild-type, *ref6-1,* and complemented lines (*TPS1::REF6;ref6-1* a, **b, and c**) with and without stratification. Details are shown in the legend of (**a**). (**c**) and (**d**) Germination phenotypes of *ref6-1* mutant embryos dissected from seeds without (**c**) and with (**d**) stratification. Embryos of wild-type, *ref6-1,* and *ref6-3* plants were manually dissected from seeds. Plots show the percentage of seedlings with root hairs and green cotyledons. Error bars indicate SD from three biological replicates (n = 90 total). The data at each time point were evaluated using one-way ANOVA, no significant differences were detected (p>0.05).

*Figure 4 continued on next page*

*Figure 4 continued*

The online version of this article includes the following figure supplement(s) for figure 4:

**Figure supplement 1.** Plots show the expression of *REF6* in the endosperm of *EPR1::REF6;ref6-1* and *TPS1::REF6;ref6-1* during germination as determined by RT-qPCR.

at 10 days after harvesting or required stratification when seeds developed under low temperature (*Figure 6—figure supplement 1*). Interestingly, the mutations in *SUVH4/5/6* only caused a delay in germination when maternally, but not when paternally inherited (*Figure 6a*), revealing a maternal effect. Since *SUVH4/5/6* genes are not regulated by genomic imprinting (*Moreno-Romero et al., 2019*; *Pignatta et al., 2014*), one possible explanation is that activity of SUVH4/5/6 in the central cell of the female gametophyte has a lasting effect until germination. Based on published expression studies, *SUVH4* and *SUVH6* are expressed in the central cell (*Belmonte et al., 2013*; *Hsieh et al., 2011*; *Wuest et al., 2010*), supporting this notion. Furthermore, we found a similar enrichment of methylated REF6 binding sites in genes marked with H3K9me2 and H3K27me3/H3K9me2 in central cells and developing endosperm (*Figure 2a* and *Figure 2—figure supplement 1*), supporting the idea that H3K9me2 and CHGm are established in the central cell. We tested this hypothesis by analyzing germination of F1 seeds generated from heterozygous *suvh4/5/6* (*suvh4/+ suvh5/+ suvh6/+*) mutants pollinated with wild-type plants. Germination of these F1 seeds was clearly delayed (*Figure 6b*), supporting the idea that SUVH4/5/6 act in the female gametophyte. We tested whether indeed CHGm has a functional role in controlling seed dormancy by testing the effect of mutations in *CMT3*. Consistently, loss of *CMT3* caused a delay in germination (*Figure 6c*) and prolonged after-ripening of *cmt3* seeds decreased primary dormancy (*Figure 6—figure supplement 2*). Interestingly, contrary to the maternal effect of *suvh4/5/6*, loss of maternal and paternal *CMT3* alleles was required to cause a delay in germination (*Figure 6d*). Based on DNA methylome data CHGm is established in the central cell before fertilization (*Figure 2a*; *Park et al., 2016*); nevertheless, the paternal *CMT3* allele was sufficient to suppress delay of germination in *cmt3*. Previous work revealed that loss of *SUVH4/5/6* causes severe depletion of both H3K9me2 and CHGm (*Stroud et al., 2014*; *Underwood et al., 2018*), while loss of *CMT3* causes depletion of CHGm but only intermediate reductions of H3K9me2 (*Stroud et al., 2014*; *Inagaki et al., 2010*). Together, this suggests that H3K9me2 is established first and recruits CHGm and that CHGm establishment after fertilization is sufficient to control seed dormancy.

## SUVH4/5/6 suppress genes with H3K27me3/H3K9me2 and CHGm on REF6-binding sites

To further test the hypothesis that H3K9me2/CHGm block the access of REF6, we analyzed transcriptome data of *suvh456* endosperm during germination. There were more than twice as many significantly up- than downregulated genes in *suvh456* compared to wild type (3,203 upregulated [$Log_2FC \geq 1$, $p<0.05$] vs 1,502 downregulated [$Log_2FC \leq -1$, $p<0.05$]; *Figure 7—source data 1*). Importantly, genes marked with H3K27me3/H3K9me2 and CHGm on REF6 binding sites were significantly enriched among upregulated, but not downregulated genes (*Figure 7a,b*), supporting the idea that REF6 targeting is blocked by H3K9me2/CHGm in the endosperm. Consistently, REF6-binding sites were significantly enriched in the gene bodies of upregulated genes marked with H3K27me3/H3K9me2/CHGm (*Figure 7c*). REF6-binding sites were not significantly enriched among the downregulated genes in *suvh456* (*Figure 7—figure supplement 1*). We also did not find significant differences among the up- and downregulated genes with single H3K9me2 (*Figure 7—figure supplement 2*), revealing that the suppressive effect of SUVH4/5/6 on gene repression connects to the presence of H3K27me3. Together, our data show that SUVH4/5/6 suppress genes with H3K27me3/H3K9me2/CHGm and that loss of *SUVH4/5/6* causes upregulation of genes through REF6-binding sites. The *ABA INSENSITIVE 3* (*ABI3*) gene was previously shown to be upregulated in *suvh45* mutants (*Zheng et al., 2012*) and was also present among the 26 upregulated genes in *suvh456* marked with H3K27me3/H3K9me2/CHGm and having REF6-binding sites (*Figure 7—figure supplement 3*). ABI3 is a B3 type transcription factor and positively regulates seed dormancy (*Liu et al., 2013*; *Shu et al., 2016*; *Tian et al., 2020*). HEAT SHOCK TRANSCRIPTION FACTOR C1 (HSFC1) was also reported to be induced in developing seeds by heat stress and suggested to activate *HEAT SHOCK PROTEIN* (*HSP*) genes that are abundant in seed maturation (*Chiu et al., 2012*). The upregulation of the two genes in the

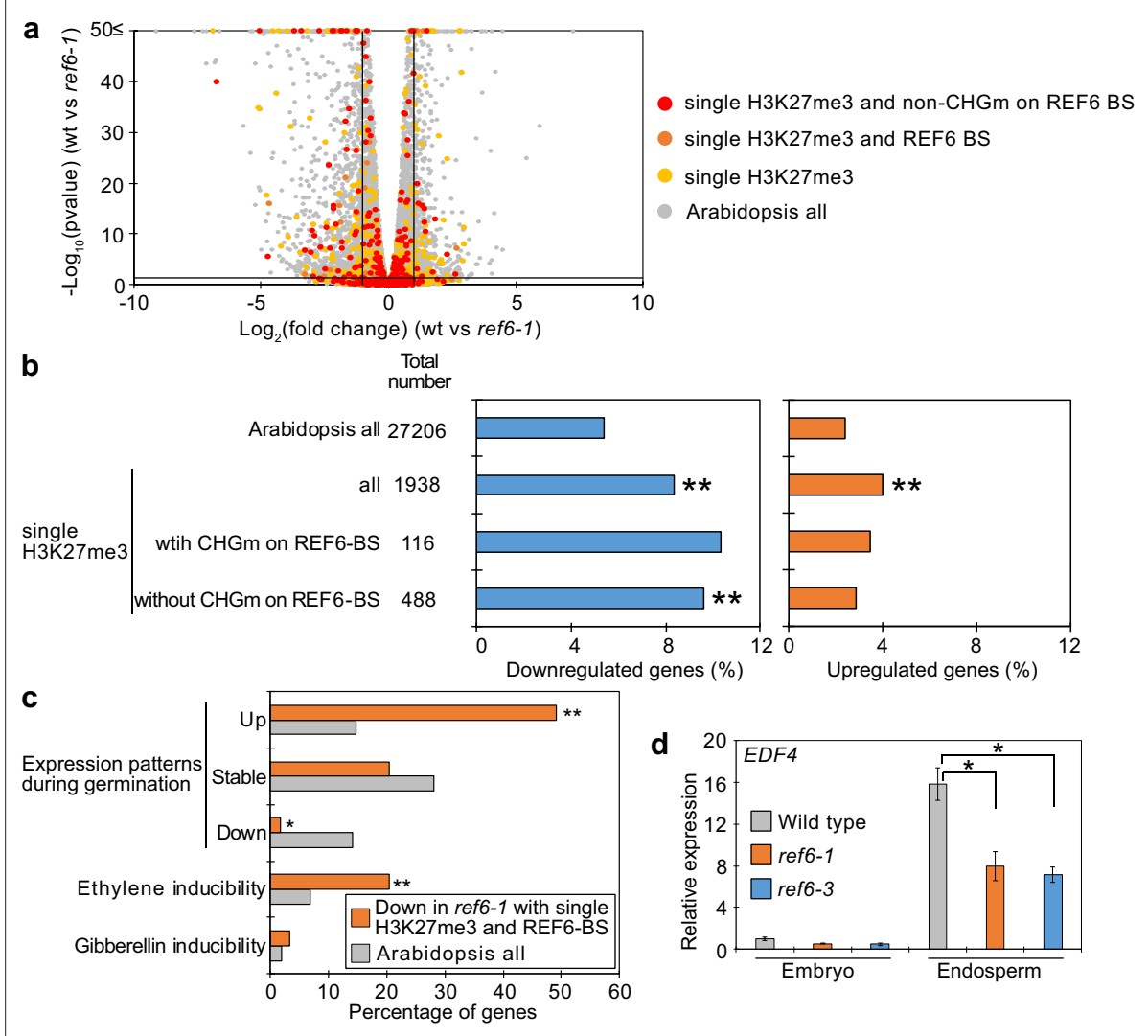

**Figure 5.** Genes marked by single H3K27me3 are activated by REF6 during germination. (**a**) Volcano plot shows differentially expressed genes with single H3K27me3 and REF6-binding sites (REF6-BS) in *ref6-1*. The color of dots corresponds to the following gene categories: red dots represent genes with single H3K27me3 and non-CHG methylation (CHGm) on REF6-BS, orange dots represent genes with single H3K27me3 and REF6-BS (excluding non-CHGm REF6-BS), yellow dots represent genes with single H3K27me3 (excluding REF6-BS), and gray dots represent *Arabidopsis* all genes (excluding genes with single H3K27me3). Vertical and horizontal lines mark thresholds of up- and downregulated genes (1≤Log2FC, Log2FC≤–1 and p<0.05). (**b**) Plots show the percentage of up- (1≤Log2FC, p<0.05) and downregulated (Log2FC≤–1, p<0.05) genes in the endosperm of *ref6-1* during germination. Compared are genes with single H3K27me3 and presence or absence of (CHGm on REF6-BS. CHGm was determined based on published data **Park et al., 2016**). Asterisks indicate significant differences compared to *Arabidopsis* all genes (*p<0.05, **p<0.01, pairwise Fisher's exact test with Benjamini–Hochberg correction). (**c**) Plot shows the expression of genes with single H3K27me3 and REF6-BS that were downregulated in endosperm of *ref6-1*. Gene expression was calculated based on tissue-specific transcriptome data in the endosperm during germination (**Dekkers et al., 2013**). Expression levels were compared between 1 and 38 hours after imbibition. Expression changes were defined as follows: upregulated genes (1≤Log2FC and p<0.05), downregulated genes (Log2FC≤–1 and p<0.05), stably expressed genes (–1≤Log2FC≤1 and p<0.05) and non-expressed genes. Percentage of ethylene and gibberellin inducible genes was calculated based on transcriptome data from seedlings (**Cao et al., 2006**; **Das et al., 2016**). Asterisks indicate significant differences compared to *Arabidopsis* all genes (*p<0.01, **p<0.001, Fisher's exact test). (**d**) Plot shows expression of *EDF4* in the endosperm of *ref6-1* during germination as determined by RT-qPCR. Embryos of wild-type, *ref6-1,* and *ref6-3* mutants were dissected after 72 hr of stratification and 24 hr of incubation under normal conditions. Error bars indicate SD from technical triplicates. Asterisks indicate significant differences compared to wild type (p<0.01, Bonferroni-corrected Student's t test).

The online version of this article includes the following source data and figure supplement(s) for figure 5:

**Source data 1.** RNA-seq data in the *ref6* mutant endosperm during germination.

**Figure supplement 1.** Transcriptome analyses of genes with single H3K9me2 in the *ref6* endosperm during germination.

*Figure 5 continued on next page*

*Figure 5 continued*

**Figure supplement 2.** Transcriptome analyses of genes with double H3K27me3/H3K9me2 in the *ref6* endosperm during germination.

**Figure supplement 3.** List of downregulated genes with single H3K27me3 and REF6-binding sites that were downregulated in *ref6-1* and expression of *CYP707A* family genes and ethylene biosynthesis genes.

**Figure supplement 4.** Analysis of the effect of exogenous chemical treatment impacting on ABA and ethylene pathways in the *ref6-1* mutant.

endosperm of *suvh456* were also confirmed by RT-qPCR (*Figure 7d*). We found that genes with single H3K27me3 were significantly enriched for downregulated genes in the *suvh456* endosperm (*Figure 7—figure supplement 4*), suggesting an indirect effect. Interestingly, we found RY binding motifs for B3 type transcription factors to be significantly enriched among downregulated genes in *suvh456* containing single H3K27me3 (*Figure 7—figure supplement 4*). This raised the hypothesis that upregulated *ABI3* in *suvh456* suppressed genes with single H3K27me3. In support of this idea, we found that genes with single H3K27me3 significantly overlapped with previously identified direct target genes of ABI3 (*Tian et al., 2020*), while genes with single H3K9me2 and double H3K27me3/H3K9me2 did not significantly overlap (*Figure 7—figure supplement 4*).

## Parentally biased expression during germination is regulated by REF6 and SUVH456

Our data obtained in this (*Figure 2*) and previous studies (*Moreno-Romero et al., 2019*; *Moreno-Romero et al., 2016*) indicate that H3K27me3, H3K9me2, and CHGm are established in the central cell and control imprinted expression in the endosperm after fertilization. We next addressed the question whether maternal-specific epigenetic marks also control parental-specific gene expression in the endosperm during germination. Using previously published parental-specific transcriptome data of the endosperm during germination (F1 seeds of maternal Col and paternal Cvi accessions) (*Piskurewicz et al., 2016*), we found that genes with triple repressive marks H3K27me3/H3K9m2/CHGm on REF6-binding sites were significantly paternally biased compared to *Arabidopsis* all genes in both dormant and non-dormant seeds (*Figure 8a*, upper blue boxes), consistent with our data showing the relationship between triple repressive marks and expression during germination (*Figure 2c*). This data suggests that triple repressive marks are associated with paternally-biased gene expression in germinating endosperm, similar as in the developing endosperm (*Moreno-Romero et al., 2019*; *Moreno-Romero et al., 2016*). In contrast, genes with single H3K27me3 and unmethylated REF6-binding sites were significantly maternally biased in dormant seeds (*Figure 8a*, lower orange boxes, left panel), but not in non-dormant seeds (*Figure 8a*, lower gray boxes, right panel). The maternally-biased expression in dormant seeds supports the idea that genes with single H3K27me3 are reactivated during germination. We observed a weak maternal bias of genes marked with single H3K27me3 already in the developing endosperm at 4 DAP (*Figure 8—figure supplement 1*), suggesting that gene activation by removing single H3K27me3 starts before germination, consistent with the observed maternal effect of *ref6* (*Figure 3b,c*). In non-dormant seeds, however, unbiased expression of genes with single H3K27me3 suggests that the paternal alleles of these genes are also activated. Indeed, this expression pattern has been previously reported for the germination-related *CP1* gene, which is maternally-biased in dormant seeds, while biallelically expressed in non-dormant seeds (*Piskurewicz et al., 2016*). We did not observe a significant parental bias of genes with single H3K9me2 (*Figure 8—figure supplement 1*).

Based on our data showing that *ref6* mainly affected genes with single H3K27me3 while *suvh4/5/6* mainly affected genes with triple repressive marks on REF6-binding sites (*Figure 5a, b* and *Figure 7a,b*), we selected genes that were likely to be regulated by REF6 and SUVH4/5/6 and analyzed the maternal expression ratios. Using the same published parental-specific transcriptome data (*Piskurewicz et al., 2016*), we found that genes with single H3K27me3 that were downregulated in *ref6* were significantly maternally biased in both dormant and non-dormant seeds (*Figure 8b*, orange boxes). Conversely, upregulated genes in *suvh456* that were marked with triple repressive marks were significantly paternally biased (*Figure 8b*, blue boxes). Nevertheless, the number of genes in each category was relatively small and further experiments should validate these conclusions. Together, this data suggests that REF6 and SUVH4/5/6 regulate genomic imprinting during germination.

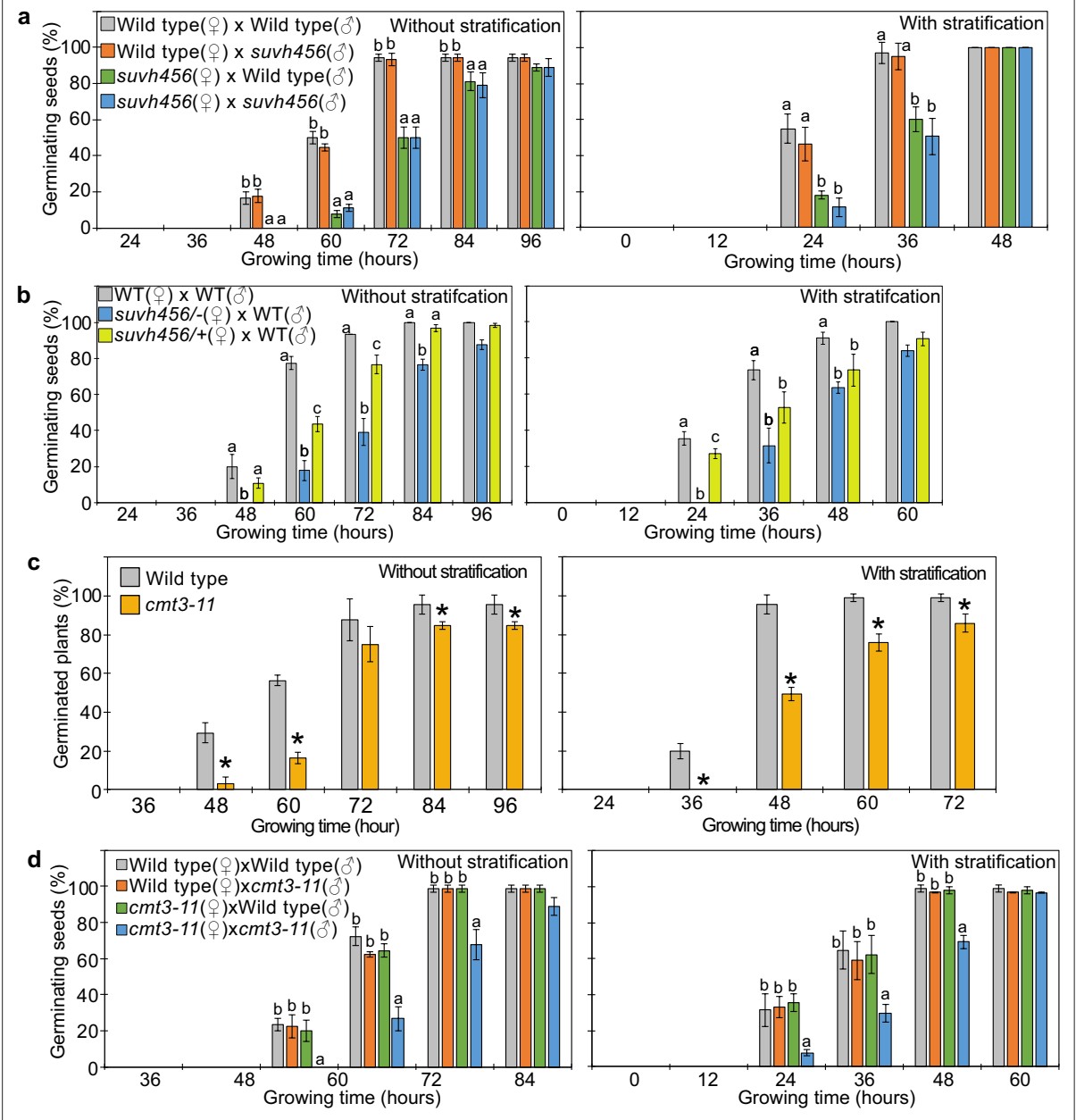

**Figure 6.** SUVH4/5/6 and CMT3 are necessary to promoter germination. (**a**) Germination phenotypes of seeds generated by reciprocal crosses between wild-type and *suvh456* triple mutants. Plots show the percentage of germinating seeds with and without stratification. Error bars indicate SD from three biological replicates (n = 120 total). The letters above the bars indicate significant differences between genotypes (p<0.05, Tukey's multiple range test). (**b**) Plots show the percentage of germinating seeds with and without stratification of wild-type (WT) and F1 seeds generated by crossing maternal heterozygous or homozygous *suvh456* and paternal wild type. Error bars indicate SD from three biological replicates (n = 90 total). Letters above bars indicate significant differences among genotypes (p<0.05, Tukey's multiple range test). (**c**) Plots show percentage of germinating seeds of wild-type and *cmt3-11* mutant with and without stratification. Error bars indicate SD from three biological replicates (n = 120 total). Asterisks indicate significant differences compared to wild type (p<0.05, Student's t test). (**d**) Germination phenotypes of seeds generated by reciprocal crosses between wild-type and *cmt3-11* mutant. Details are shown in the legend of (**a**).

The online version of this article includes the following figure supplement(s) for figure 6:

**Figure supplement 1.** Seed dormant phenotypes of *suvh456* mutants.

**Figure supplement 2.** Seed dormant phenotypes of *cmt3-11* mutant.

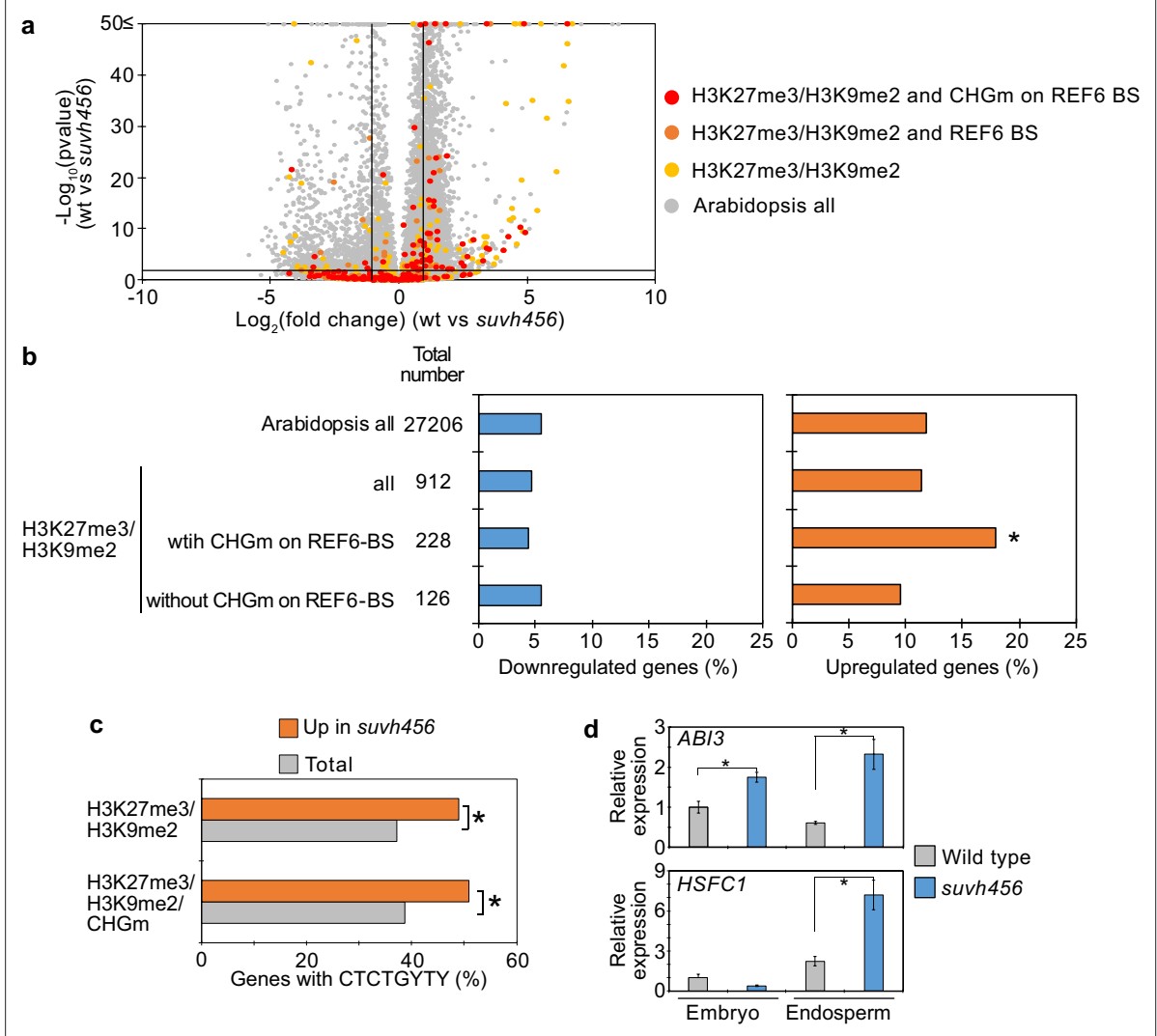

**Figure 7.** SUVH4/5/6 are required for the suppression of genes with H3K27me3/H3K9me2 and CHG methylation (CHGm) on REF6-binding sites (REF6-BS). (**a**) Volcano plot shows differentially expressed genes with double H3K27me3/H3K9me2 and CHGm on REF6-BS in *suvh456*. The color of dots corresponds to the following gene categories: red dots represent genes with double H3K27me3/H3K9me2 and CHGm on REF6-BS, orange dots represent genes with double H3K27me3/H3K9me2 and REF6-BS (excluding CHGm REF6-BS), yellow dots represent genes with double H3K27me3/H3K9me2 (excluding REF6-BS), and gray dots represent *Arabidopsis* all genes (excluding genes with H3K27me3/H3K9me2). Vertical and horizontal lines mark thresholds of up- and downregulated genes are shown (1≦Log$_2$FC, Log$_2$FC≦–1, p<0.05). (**b**) Plots show the percentage of up-(1≦Log$_2$FC, p<0.05) and downregulated (Log$_2$FC≦–1, p<0.05) genes in the endosperm of *suvh456* during germination. Compared are genes with double H3K27me3/H3K9me2 and presence or absence of CHGm on REF6-BS. CHGm was determined based on published data (**Park et al., 2016**). Asterisks indicate significant differences compared to *Arabidopsis* all genes (p<0.05, pairwise Fisher's exact test with Benjamini–Hochberg correction). (**c**) Plot shows the enrichment of REF6-BS among upregulated (1≦Log$_2$FC, p<0.05) genes in the endosperm of *suvh456* during germination. Compared are genes with the indicated combinations of histone modifications and presence (>0.04) or absence (≦0.04) of CHGm. CHGm levels were determined based on published data (**Park et al., 2016**). Asterisks indicate significant differences between two categories (p<0.05, Fisher's exact test). (**d**) Plots show the expression of *ABI3* and *HSFC1* in the endosperm of *suvh456* during germination as determined by RT-qPCR. Embryos of wild type and the *suvh456* triple mutant were dissected after 72 hr of stratification and 24 hr of incubation under normal conditions. Error bars indicate SD from technical triplicates. Asterisks indicate significant differences between wild type and *suvh456* (p<0.01, Student's t test).

The online version of this article includes the following source data and figure supplement(s) for figure 7:

**Source data 1.** RNA-seq data in the *suvh456* mutant endosperm during germination.

**Figure supplement 1.** Enrichment of REF6-binding sites (REF6-BS) among downregulated genes in the endosperm of *suvh456* during germination.

**Figure supplement 2.** Transcriptome analyses of genes with single H3K9me2 in *suvh456* mutant endosperm during germination.

**Figure supplement 3.** List of upregulated genes (1≦Log$_2$FC, p<0.05) with H3K27me3, H3K9me2, CHG methylation, and REF6-binding sites in *suvh456*

*Figure 7 continued on next page*

*Figure 7 continued*

endosperm of during germination.

**Figure supplement 4.** Transcriptome analyses of genes with single H3K27me3 in *suvh456* mutant endosperm during germination.

**Figure supplement 5.** Possible effect of upregulated *ABI3* on the suppression of genes with single H3K27me3 in *suvh456*.

The maternally biased expression of genes with single H3K27me3 suggests suppression of the paternal alleles by an epigenetic mechanism. Maternally biased gene expression during germination was previously reported to be mediated by the non-canonical RNA-directed DNA methylation (RdDM) pathway targeting the paternal alleles (*Iwasaki et al., 2019*). We therefore analyzed DNA methylation levels in genic as well as 2 kb up- and downstream regions on the paternal alleles of genes with single H3K27me3 on the maternal alleles using previously published data (*Ibarra et al., 2012*; *Moreno-Romero et al., 2016*). We found that paternal CHHm levels in downstream regions of genes with maternal H3K27me3 and in gene bodies of genes with maternal H3K27me3 and unmethylated REF6-binding sites were significantly higher compared to *Arabidopsis* all genes in 7–8DAP endosperm (*Figure 8—figure supplement 2*). CHGm levels in downstream regions were significantly higher in 4DAP endosperm, but this trend was not detected at 7–8DAP (*Figure 8—figure supplement 3*). Levels of CG methylation (CGm) were marginally increased in gene bodies of genes with maternal H3K27me3 and unmethylated REF6-binding sites in 4DAP and 7–8DAP endosperm (*Figure 8—figure supplement 4*). This data suggests that CHHm of genes with maternal H3K27me3 and unmethylated REF6-binding sites can be one possible factor to establish maternally biased expression of genes with single H3K27me3 in dormant seeds. However, detailed parental-specific DNA methylome data in dormant and non-dormant seeds are necessary to validate this conclusion.

## Discussion

In this study, we uncovered the molecular function and physiological role of the endosperm-specific triple modification of the maternal genome with H3K27me3/H3K9me2 and CHGm. We found that genes marked by only H3K27me3 became reactivated during seed germination, a process that depends on the H3K27me3 demethylase REF6. In contrast, genes marked by triple repressive modifications remained silenced during germination because REF6 binding is repelled by H3K9me2/CHGm, consistent with previous work (*Qiu et al., 2019*). Supporting the functional requirement of REF6 to initiate germination, loss of *REF6* caused higher seed dormancy (*Figure 3*, *Figure 3—figure supplement 2*; *Chen et al., 2020*), connected with decreased expression of many germination-inducible genes. Loss of *SUVH4/5/6* also increased dormancy (*Figure 6—figure supplement 1*; *Zheng et al., 2012*), which was associated with increased expression of potential REF6 target genes. Since H3K9me2 acts in a feedback loop with CHGm (*Du et al., 2012*; *Jackson et al., 2002*; *Lindroth et al., 2001*; *Malagnac et al., 2002*), loss of *SUVH4/5/6* is expected to expose unmethylated binding sites for REF6 that can then be activated by REF6. Consistent with this model, loss of *CMT3* also caused increased dormancy (*Figure 6b*, *Figure 6—figure supplement 2*). Interestingly, among those upregulated SUVH4/5/6 targets was the positive regulator of seed dormancy *ABI3* (*Liu et al., 2013*), providing a possible explanation for the increased dormancy in *suvh4/5/6*. Consistent with this notion we found that down-regulated genes with single H3K27me3 were enriched for ABI3 binding motifs. Nevertheless, the small fraction of germination-inducible genes with triple repressive marks suggests possible additional mechanisms removing triple epigenetic marks.

Our work furthermore revealed the connection between maternal-specific repressive marks, gene imprinting, and seed dormancy. Imprinted genes were previously shown to play a role in regulating seed germination and were connected to the maternal inheritance of seed dormancy levels (*Iwasaki et al., 2019*; *Piskurewicz et al., 2016*). The non-canonical RdDM pathway was found to target and silence the paternal alleles of the dormancy-related gene *ALN1* (*Iwasaki et al., 2019*). Our data suggests that in addition to RdDM, maternally biased expression of dormancy-related genes also requires REF6, which activates maternal H3K27me3 marked genes. We propose that both mechanisms together ensure the maternal control of seed dormancy; while REF6 regulates the time of activation of certain genes, RdDM-mediated CHHm on the paternal alleles prevents their activation. Interestingly, the timing of REF6 action probably differs for distinct genes. While some genes, like *ALN1, CYP707A1*, and *CYP707A3* (*Chen et al., 2020*) are likely activated in the central cell of the female

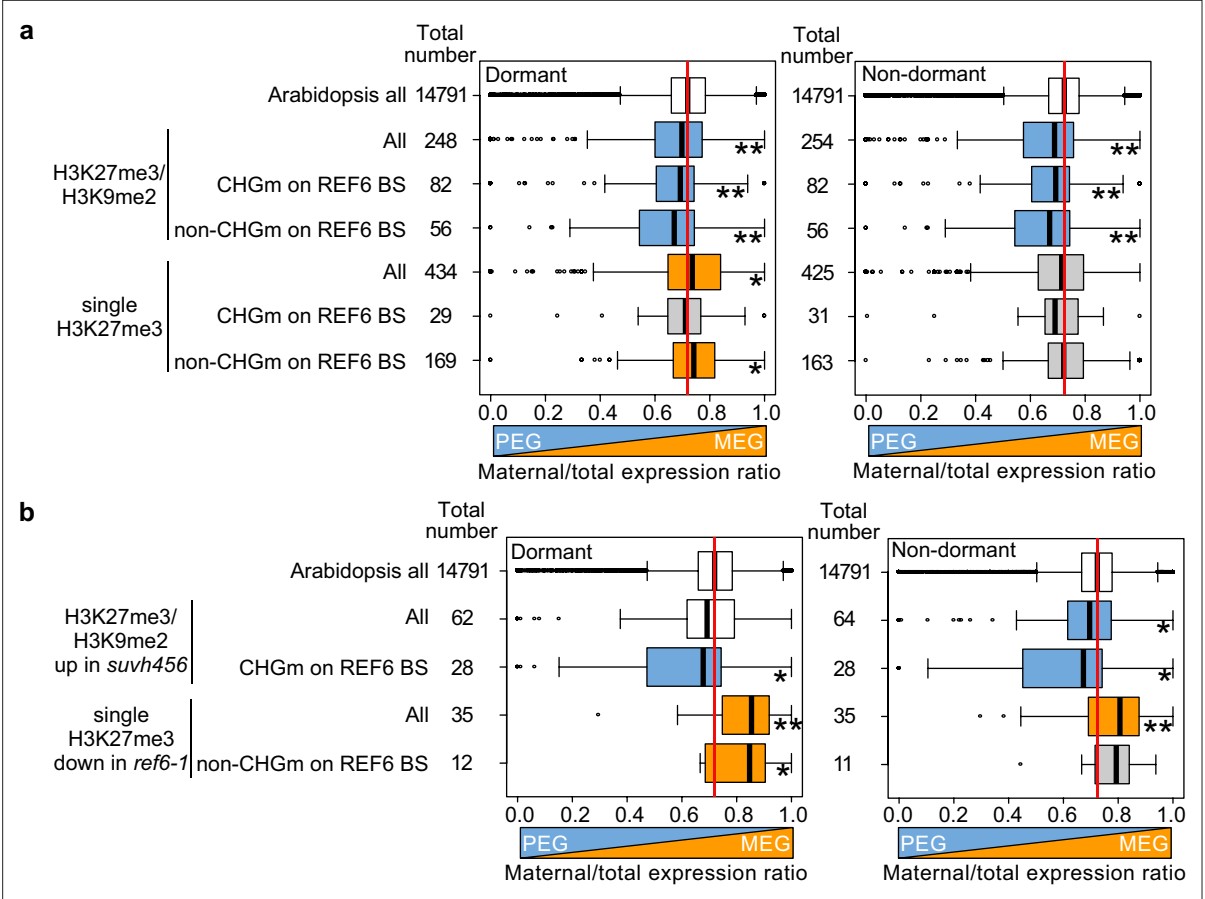

**Figure 8.** Maternal- and paternal-specific epigenetic marks regulate parental-specific gene expression in the endosperm during germination. (**a**) Box plots show the mean values of maternal to total reads of genes expressed in the endosperm in dormant and non-dormant seeds. Compared are *Arabidopsis* all genes, all genes with double H3K27me3/H3K9me2 and single H3K27me3 depending on the presence or absence of CHG methylation (CHGm) on REF6-binding sites (REF6-BS) based on published data (*Park et al., 2016*). Parental-specific transcriptome data in the endosperm during germination from F1 seeds (maternal Col × paternal Cvi) were previously published (*Piskurewicz et al., 2016*). Boxes show medians and the interquartile range, and error bars show full-range excluding outliers. Numbers of genes in each category are shown beside the boxes. Asterisks indicates significant differences compared to *Arabidopsis* all genes (*p<0.05, **p<0.01, pairwise Wilcoxon test with Benjamini–Hochberg correction). Significantly paternally and maternally biased groups are shown in blue and orange, respectively. (**b**) Box plots show mean values of maternal to total reads in the endosperm of dormant and non-dormant seeds of *Arabidopsis* all genes, upregulated genes with double H3K27me3/H3K9me2 in *suvh456*, and downregulated genes with single H3K27me3 in *ref6-1*. Details are shown in the legend of (**a**).

The online version of this article includes the following figure supplement(s) for figure 8:

**Figure supplement 1.** Effects of maternal-specific epigenetic marks on parental-specific gene expression profiles in the endosperm during development.

**Figure supplement 2.** Levels of paternal CHH methylation at 4DAP (**a**) and 7-8DAP (**b**) on genes marked by H3K27me3 on the maternal alleles and with methylated or unmethylated REF6-BS within gene bodies and 2 kb up- and downstream regions.

**Figure supplement 3.** Levels of paternal CHG methylation at 4DAP (**a**) and 7-8DAP (**b**) on genes marked by H3K27me3 on the maternal alleles and with methylated or unmethylated REF6-BS within gene bodies and 2 kb up- and downstream regions.

**Figure supplement 4.** Levels of paternal CG methylation at 4DAP (**a**) and 7-8DAP (**b**) on genes marked by H3K27me3 on the maternal alleles and with methylated or unmethylated REF6-BS within gene bodies and 2 kb up- and downstream regions.

**Figure supplement 5.** Model depicting the relationship between epigenetic patterns in the endosperm during seed development and gene regulatory mechanisms in the endosperm during seed germination.

gametophyte and therefore lack detactable H3K27me3 in the endosperm, other genes are activated during endoperm development. To the latter category belongs the previously reported gene *CP1*, which is maternally biased in dormant seeds and biallelically expressed in non-dormant seeds (*Piskure-wicz et al., 2016*) and marked with single H3K27me3 on the maternal alleles (*Moreno-Romero et al.,*

*2016*). In contrast to genes with single H3K27me3, genes with triple repressive marks on maternal alleles are not targeted by REF6, causing the maternal alleles to be silenced and the paternal alleles to be active. Genes with triple repressive marks on REF6-binding sites were upregulated *in suvh4/5/6* and were paternally biased during germination. Although more detailed parental-specific DNA methylome analyses in germinating endosperm are required to precisely determine the molecular mechanism repressing the paternal alleles of genes with maternal H3K27me3 during germination, this data supports the model that the presence of single H3K27me3 and triple repressive marks determines whether genes are maternally or paternally biased during germination, respectively (*Figure 8—figure supplement 5*).

The *SUVH4/5/6* genes are not imprinted (*Hsieh et al., 2011*; *Moreno-Romero et al., 2019*; *Pignatta et al., 2014*); however, the delay of germination in *suvh4/5/6* was under maternal control (*Figure 6a*). This indicates that maternal-specific triple repressive marks are established before fertilization in the central cell and maintained after fertilization, consistent with previous data (*Moreno-Romero et al., 2019*). This suggests that maternal-specific triple epigenetic marks of H3K27me3/H3K9me2 and CHGm contribute to maternally controlled seed dormancy (*Iwasaki et al., 2019*; *Piskurewicz et al., 2016*). Likewise, also *REF6* is not imprinted (*Hsieh et al., 2011*; *Pignatta et al., 2014*), but *ref6* had a maternal effect (*Figure 3b*), supporting the notion that derepression of some dormancy-controlling genes starts in the central cell of the female gametophyte. Nevertheless, the fact that some REF6 targets have detectable levels of H3K27me3 in the endosperm and since we could partially complement the *ref6* mutant by expressing *REF6* in the micropylar domain of the endosperm (*Figure 5a*), strongly suggests that REF6 also acts after fertilization. In support of this view, homozygous *ref6* mutants had a stronger delay in germination than maternal *ref6* mutants (*Figure 3b*).

Our data furthermore uncovered a relationship among REF6, ethylene, and ABA. Consistent with previous work showing that loss of *ABA2* does not completely suppress the delayed germination of *ref6* after stratification (*Chen et al., 2020*), we found that the ABA biosynthesis inhibitor fluridone was not sufficient to suppress delayed germination phenotypes of *ref6* (*Figure 5—figure supplement 4*). Similarly, the ethylene precursor ACC caused only a partial suppression of the germination delay. However, the combined treatment with fluridone and ACC completely suppressed the *ref6* germination phenotype (*Figure 5—figure supplement 4*), supporting our and previous data revealing that REF6 is required to activate ABA catabolism and ethylene signaling through removal of H3K27me3. Since the activation of the ABA catabolism genes *CYP707A1* and *CYC707A3* likely occurs in the central cell while activation of ethylene signaling genes occurs during endosperm development, we propose that there is a two-step maternal control of germination by REF6. The first step occurs in the central cell and leads to the suppression of ABA signaling, while the second step occurs in the endosperm and leads to the activation of ethylene signaling. This two-step control of germination may ensure the best timing of germination. Understanding by which mechanism the timing of REF6 action is regulated will be subject of future investigations.

Previous data showed that loss of FIS-PRC2 function causes upregulation of genes with both H3K27me3 and H3K9me2 (*Hsieh et al., 2011*; *Moreno-Romero et al., 2016*), suggesting that establishment of H3K9me2 depends on H3K27me3. This idea was substantiated by data showing that deficiency of FIS-PRC2 in the central cell causes depletion of CHGm (*Moreno-Romero et al., 2019*), pointing that H3K9me2/CHGm depend on FIS-PRC2. Also data shown in this study are consistent with this scenario; we demonstrate that loss of H3K9me2/CHGm only caused an effect if H3K27me3 was removed by REF6, revealing that H3K27me3 is the primary repressive mark. Additionally, we found that maternal loss of *SUVH4/5/6* was sufficient to cause delayed gemination (*Figure 6a*), while that of *CMT3* was not (*Figure 6c*). One possible explanation is that the effects caused by loss of SUVH4/5/6 and CMT3 function differ. While H3K9me2 and CHGm are severely depleted in *suvh4/5/6*; loss of *CMT3* mainly affects CHGm with only intermediate effects on H3K9me2 (*Stroud et al., 2014*; *Underwood et al., 2018*). Thus, genes with H3K27me3/H3K9me2 can likely be suppressed by CMT3 after fertilization.

A possible mechanism is that the FIS2-PRC2 complex or H3K27me3 recruits SUVH4/5/6 and that H3K9me2 recruits CMT3, suggesting the presence of a molecular mechanism that distinguishes target genes to be marked by single H3K27me3 or triple marks of H3K27me3/H3K9me2 and CHGm. To identify this distinguishing mechanism remains an important future endeavor.

Our data bear striking similarities to maternal H3K27me3-dependent non-canonical imprinting reported in mammals (*Chen et al., 2019*; *Hanna et al., 2019*; *Inoue et al., 2017a*; *Inoue et al., 2017b*). In non-canonical imprinting, maternal-specific H3K27me3 inherited from oocytes serves as the primary epigenetic mark for imprinted paternal-specific expression. Most of this non-canonical imprinting is transient and restricted to preimplantation embryos, with only some genes important for placental development remain imprinted in extraembryonic cells (*Inoue et al., 2017a*). Maintenance of non-canonical imprinting is mediated by DNA methylation, which is specifically applied on the maternal alleles of previously H3K27me3 marked regions (*Chen et al., 2019*; *Hanna et al., 2019*). Thus, similar to the findings reported in this study, genes regulated by non-canonical imprinting in mammals are transiently imprinted by inheriting maternal-specific H3K27me3 and only few genes remain imprinting by establishing DNA methylation as secondary imprint. Genomic imprinting has independently evolved in flowering plants and mammals; however, rather remarkably, similar molecular mechanisms have been coopted to regulate imprinted genes in the accessory, nutrient-transferring tissues.

The combined presence of H3K27me3 with the heterochromatic modification H3K9me3 was also reported to occur on TEs and repeat-rich regions in the protozoa *Tetrahymena* and *Paramecium* (*Frapporti et al., 2019*; *Liu et al., 2007*; *Zhao et al., 2019*) and the fungus *Podospora anserine* (*Carlier et al., 2020*). In *P. anserine*, the presence of double repressive modifications is restricted to TEs and repeat-rich regions, while H3K27me3 alone is located in genic regions, indicating a similar molecular distinction of constitutively silenced regions marked by double modifications from regions that require activation during specific developmental stages marked by H3K27me3 alone. It is tempting to speculate that the co-occurrence of H3K9me3 and H3K27me3 may prevent H3K27me3 demethylases to target and activate those regions, similar to the findings reported in this study.

In summary, in this study, we uncovered a new molecular mechanism leading to maternal-specific dormancy control that distinguishes genes marked by either H3K27me3 alone from genes marked by the generally antagonistic repressive epigenetic modifications, H3K27me3, H3K9me2, and CHGm. We uncovered how the H3K27me3 demethylase REF6 controls release of dormancy and revealed the underlying mechanism for the H3K9me2 methylases SUVH4/5/6 in controlling dormancy and propose that these regulators form a regulatory network through ABA and ethylene pathways. Our data provide new insights into how maternal-specific epigenetic modifications control dormancy and thus ensure timely germination.

## Materials and methods
### Plant material and growing conditions

All mutants used in this study are in the Col background. The T-DNA insertion lines of *ref6-1* (SALK_001018) (*Cui et al., 2016*), *ref6-3* (SAIL_747_A07) (*Cui et al., 2016*), *elf6-3* (SALK_074694) (*Tao et al., 2017*), *elf6-4* (SAIL_371_D08) (*Tao et al., 2017*), and an ethylmethane sulfonate-mutated line of *cmt3-11* (*Lindroth et al., 2001*) were previous described. The T-DNA insertions were confirmed by amplifying the left border-flanking regions of the genomic DNA using the primer set given in *Supplementary file 1*. The *suvh456* mutant was kindly provided by Judith Bender. The *elf6-3 ref6^C* double and *elf6-3 ref6^C jmj13^G* triple mutants were kindly provided by Kerstin Kaufmann (*Yan et al., 2018*). The *cyp707a2-1* and *cyp707a2-2* mutants were kindly provided by Eiji Nambara (*Kushiro et al., 2004*). All plants were grown on half-strength Murashige and Skoog agar medium containing 1 % sucrose or on soil under long-day conditions (16 hr light/8 hr darkness) at 21 °C and a light intensity of 150 µE. Seeds were surface sterilized (5 % sodium hypochlorite) before sawing on medium and 14-day-old plants were transferred into soil. For germinating assays, seeds from dry siliques were harvested on the same day 3 weeks after pollination and after-ripened at room temperature for 10 days unless otherwise indicated. Transgenic plants were generated through the floral dip method by using *Agrogacterium tumefaciens* GM3101 as described previously (*Clough and Bent, 1998*).

### Germinating assays

For germinating assays, seeds from 20 siliques were sterilized and grown on half-strength Murashige and Skoog medium with or without stratification at 4 °C for 3 days. To analyze the effects of chemicals on germination, 1-aminocyclopropanecarboxylic acid (ACC) (Sigma-Aldrich) and fluridone (Merck

Chemicals and Life Science AB) were added to half-strength Murashige and Skoog medium. For seed dormancy assays, plants were grown under normal conditions or under lower temperature (10 °C). The sterilized seeds were sown on minimal media (0.9 % agar) and grown under normal conditions above with or without stratification. For growth analysis of dissected embryos, seeds were sterilized and manually dissected after treatment of stratification as previously described (*Piskurewicz et al., 2016*). Root protrusion, root hair formation, and cotyledon greening were observed under a microscope (Leica, EZ4) at different times during germination, and photographs were taken by a high-resolution digital camera (Leica, M205 FA). Experiments were done in biological replicates, as indicated.

## RNA preparation and qRT-PCR

For gene expression analysis, embryos and endosperm were manually dissected from seeds that were imbibed for 24 hr as previously described (*Endo et al., 2012*; *Piskurewicz et al., 2016*). Total RNA from embryos and endosperm was extracted using MagMAX Plant RNA Isolation Kit (Thermo Fisher Scientific) according to the supplier's instructions for RNA extraction from plant seeds. For quantitative RT-PCR, cDNA was synthesized from total RNA using the Thermo Scientific RevertAid First-Strand cDNA Synthesis Kit according to the supplier's instructions. Quantitative RT-PCR analyses were performed using a BIO RAD CFX Connect Real-Time PCR Detection System. HOT FIREPol EvaGreen qPCR Mix Plus (ROX) (Solis BioDyne) was used for the reactions. The primers used for quantitative RT-PCR are listed in *Supplementary file 1*. Triplicate measurements were generated for each cDNA sample, and the obtained values were normalized to *ACT2*.

RNA sequencing and data analysis mRNA was isolated from total RNA using the NEBNext Poly(A) mRNA Magnetic Isolation Kit (NEB) and cDNA libraries were constructed using the NEBNext Ultra II RNA Library Prep Kit for Illumina (NEB). Libraries were sequenced on Illumina NovaSeq platforms as paired-end through Novogene mRNA sequencing services. For each replicate, the 150 bp long reads were trimmed by removing the 20 bp from the 5' end and 30 bp from the 3' and mapped in single-end mode to the *Arabidopsis* (TAIR10) genome, masked for rRNA genes, using TopHat v2.1.0 (*Trapnell et al., 2009*) (parameters adjusted as -g 1 -a 10 -i 40 -I 5000 F 0 r 130). Mapped reads were counted and normalized to reads per kilobase per million mapped reads for chromosomal protein coding genes using GFOLD (*Feng et al., 2012*). Differentially regulated genes across the three replicates of each condition were detected using DESeq (v. 1.12.0) (*Anders and Huber, 2010*) and in R (v. 3.3.2) (*R Development Core Team, 2016*). Only changes in expression with a FDR of <0.05 were considered statistically significant. Overrepresentation analysis of hexamers on the promoters of sets of genes by RNA-sequencing data were performed as previously described (*Maruyama et al., 2012*) using 1 kb upstream sequences from the translational start sites obtained through Phytozome (version 12; https://phytozome.jgi.doe.gov/pz/portal.html).

## Data analysis

Published data of maternal-specific histone modifications of H3K27me3 and H3K9me2 in the developing endosperm derived from crosses of maternal Col and paternal L*er* accessions (*Moreno-Romero et al., 2019*) were used to identify three groups of genes with H3K27me3, H3K9me2, and H3K27me3/ H3K9me2. We used CHG methylation scores on gene bodies in central cells (*Park et al., 2016*) as defined in a previous study (*Moreno-Romero et al., 2019*). The time-course of expression profiles in the endosperm during germination was determined using previously published data of tissues containing micropylar and chalazal endosperm (*Dekkers et al., 2013*). To analyze the genomic loci containing CHG, CHH, and CG methylation, published DNA methylation data of central cells (*Park et al., 2016*) and the developing endosperm at 4 days (*Moreno-Romero et al., 2016*) and 7–8 days (*Ibarra et al., 2012*) after pollination were reanalyzed. For the analyses of methylated sites on REF6-binding sites, methylated CHG sites were defined as those containing one or more methylated cytosines among at least five reads in both replicates. For the analyses of parental-specific methylation, numbers of methylated and unmethylated cytosines of two replicates were pooled and methylated sites were defined as those containing one or more methylated cytosines among at least five reads. To analyze parental-specific gene expression patterns in the endosperm during germination, published data of endosperm-specific transcriptomes in F1 seeds generated by reciprocal crosses of Col and Cvi (*Piskurewicz et al., 2016*) was used. Maternal expression ratios were calculated as the maternal read counts to total read counts (maternal-specific read counts/total read counts).

## Protein sequence alignment and gene ontology analysis

The peptide sequences of REF6, ELF6, and JMJ13 were obtained from Phytozome (Phytozome v12, https://phytozome.jgi.doe.gov/pz/portal.html). Alignment of the proteins was performed using Snap-Gene and ClustalW and the result of the alignment was manually adjusted to detect conserved amino acids. Gene Ontology (GO) analyses were performed using the public GO database agriGO (Version 2.0) (*Tian et al., 2017*) and processed through REVIGO using p values (*Supek et al., 2011*).

## Plasmid constructions

To generate the constructs for complementation assays, the pB7WG vector (https://gatewayvectors.vib.be/) was digested with *Eco*32I and ligated to remove the cassette of the LR reaction. The promoter regions of *EPR1* or *TPS1* were inserted into the *Kpn*I and *Xba*I sites of the vector, and the *REF6* coding sequence was inserted into the *Aat*II and *Eco*32I sites using the In-Fusion HD Cloning Kit (TaKaRa).

## Acknowledgements

We thank Cecilia Wärdig for technical support and Katarzyna Dziasek for helpful comments on the manuscript. This research was supported by grants from the Swedish Research Council VR (2017–04119, to CK), a grant from the Knut and Alice Wallenberg Foundation (2018–0206, to CK), a grant from the Göran Gustafsson Foundation for Research in Natural Sciences and Medicine (to CK), and a fellowship grant from the HFSP (to HS).

## Additional information

### Funding

| Funder | Grant reference number | Author |
|---|---|---|
| Vetenskapsrådet | 2017-04119 | Claudia Köhler |
| Knut och Alice Wallenbergs Stiftelse | 2018-0206 | Claudia Köhler |
| Goran Gustafsson Foundation for Research in Natural Sciences and Medicine | | Claudia Köhler |
| Human Frontier Science Program | LT000162/2018-L | Hikaru Sato |

The funders had no role in study design, data collection and interpretation, or the decision to submit the work for publication.

### Author contributions

Hikaru Sato, Conceptualization, Data curation, Formal analysis, Funding acquisition, Investigation, Methodology, Project administration, Resources, Software, Validation, Visualization, Writing - original draft, Writing - review and editing; Juan Santos-González, Data curation, Formal analysis, Investigation, Methodology, Software, Validation, Visualization; Claudia Köhler, Conceptualization, Data curation, Funding acquisition, Methodology, Project administration, Resources, Supervision, Validation, Visualization, Writing - review and editing

### Author ORCIDs

Hikaru Sato http://orcid.org/0000-0001-7628-0414
Juan Santos-González http://orcid.org/0000-0002-8712-9776
Claudia Köhler http://orcid.org/0000-0002-2619-4857

### Decision letter and Author response

Decision letter https://doi.org/10.7554/64593.sa1
Author response https://doi.org/10.7554/64593.sa2

## Additional files

### Supplementary files
• Supplementary file 1. Primers used in this study.

• Transparent reporting form

### Data availability
All data generated in this study were deposited to the Gene Expression Omnibus and are available under the accession number GSE158907.

The following dataset was generated:

| Author(s) | Year | Dataset title | Dataset URL | Database and Identifier |
|---|---|---|---|---|
| Köhler C | 2021 | Combinations of maternal-specific repressive epigenetic marks in the endosperm control seed dormancy | http://www.ncbi.nlm.nih.gov/geo/query/acc.cgi?acc=GSE158907 | NCBI Gene Expression Omnibus, GSE158907 |

The following previously published datasets were used:

| Author(s) | Year | Dataset title | Dataset URL | Database and Identifier |
|---|---|---|---|---|
| Moreno-Romero J, Jiang H, Santos-González J, Köhler C | 2016 | Parental epigenetic asymmetry of PRC2-mediated histone modifications in the Arabidopsis endosperm | http://www.ncbi.nlm.nih.gov/geo/query/acc.cgi?acc=GSE66585 | NCBI Gene Expression Omnibus, GSE66585 |
| Park K, Kim MY, Vickers M Park | 2016 | DNA demethylation is initiated in the central cells of Arabidopsis and rice | http://www.ncbi.nlm.nih.gov/geo/query/acc.cgi?acc=GSE89789 | NCBI Gene Expression Omnibus, GSE89789 |
| Dekkers BJ, Pearce SP, van Bolderen-Veldkamp M, Marshall A, Bassel GW, King JR, Wood AT, Müller K, Widera P, Gilbert J, Krasnogor N, Leubner-Metzger G, Holdsworth MJ, Bentsink L | 2013 | *Arabidopsis thaliana* seed germination timecourse | http://www.ncbi.nlm.nih.gov/geo/query/acc.cgi?acc=GSE41212 | NCBI Gene Expression Omnibus, GSE41212 |
| Ibarra CA, Feng X, Schoft VK, Hsieh TF, Uzawa R, Rodrigues JA, Zemach A, Chumak N, Machlicova A, Nishimura T, Rojas D, Fischer RL, Tamaru H, Zilberman D | 2012 | Active DNA demethylation in plant companion cells reinforces transposon methylation in gametes | http://www.ncbi.nlm.nih.gov/geo/query/acc.cgi?acc=GSE38935 | NCBI Gene Expression Omnibus, GSE38935 |

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
