## [Decision Letter]

**Acceptance summary:**

Sato et al. describe the role of histone modifications and DNA methylation in the control of seed dormancy in Arabidopsis. The authors show that DNA methylation associated with a specific histone modification (H3K9me2) prevents the REF6 enzyme from removing another histone modification (H3K27me3). Genes marked with both histone modifications are therefore resistant to activation, and this is important for gene regulation in the endosperm during seed

**Decision letter after peer review:**

Thank you for submitting your article "Combinations of maternal-specific repressive epigenetic marks in the endosperm control seed dormancy" for consideration by *eLife*. Your article has been reviewed by 2 peer reviewers, and the evaluation has been overseen by Daniel Zilberman as the Reviewing Editor and Jürgen Kleine-Vehn as the Senior Editor. The following individuals involved in review of your submission have agreed to reveal their identity: Steven Penfield (Reviewer #2).

The reviewers have discussed the reviews with one another and the Reviewing Editor has drafted this decision to help you prepare a revised submission. We are also appending Steven Penfield's original review because we feel it may be helpful in drafting a revised manuscript, but only the revisions indicated in this letter need to be directly addressed.

Summary:

Sato et al. describe the role of histone marks and DNA methylation in the control of seed dormancy in Arabidopsis. This paper will be of interest to scientists working on epigenetics, seed development and germination. Most of the claims are supported by data and the authors have generated many genomic datasets and knowledge that could be useful for the study of seed dormancy in other plant species. The authors show that the REF6/H3K27me3 suvh456/H3K9me2 interaction is important for gene regulation in the endosperm during germination. The authors also investigate the role of REF6 in germination responses in seeds. This aspect of the study is intriguing, but less strongly supported and is in apparent disagreement with recently published conclusions.

Essential revisions:

1. The reviewers and editors felt that the most important revisions concern the narrative that REF6 and SUVH4-6 control dormancy via ethylene responses. The mechanism by which REF6 regulates dormancy is an essential component of the manuscript. However, the presented mechanism is not convincingly supported by the supplied data and differs from the conclusions of a recently published paper (Chen et al., Plant Physiology 2020), which implicates ABA catabolism during seed development. The authors should establish the relationship between REF6, ethylene and ABA signalling, and ideally reconcile their findings with those of Chen et al., 2020.

2. Related to the above point, seed dormancy should be assessed in the absence of dormancy-breaking stimuli, especially nitrate in the MS medium. Sowing freshly harvested seed on minimal media would result in higher quality data and be the normal approach in the field, showing that seed lots indeed show dormancy (germination <50% is better). Then the authors would clearly be able to show that dormancy rather than germination is affected.

3. The authors are not always clear on the mutant allele used in different experiments. A major caveat is that the ref6-1 and ref6-3 mutants used in germination experiments (Figure 3, Figure 3¬—figure supplement 2 and Figure 3-—figure supplement 3) and most of the molecular analyses are not a full knockout (null) mutants (Yan et al., 2019 and Antunez et al., 2020). The use of partial-loss-of-function REF6 alleles could be problematic, in particular for germination assays with seeds from reciprocal crosses. To address this issue, authors should use a loss-of-function mutant line ref6C, which they have used to generate the triple elf6-3 ref6C jmj13G mutant.

Similarly, the interpretations arising from the endosperm transcriptome analysis with ref6-1 mutant plants should be carefully considered in their conclusions.

4. The interpretation that suvh4/5/6 cause a maternal effect in germination due to the activity of these histone methylases in the central cell is not fully supported. Genetic analyses would be required to distinguish sporophytic and gametophytic effects of these mutants on seed germination. Similarly, the parent-of-origin effects on germination arising in the cmt3 mutant are far from clear. The role of suvh4/5/6 and cmt3 in parent-of-origin effects on germination should be critically assessed and described in the manuscript accordingly.

5. The expression of REF6 in the micropylar endosperm and embryo using EPR1::REF6 and TPS1::REF6 should be confirmed experimentally (eg. RT-PCR with dissected tissues).

6. The strong claims about the effects of CHGm on REF6 should be moderated. The model communicated by the authors involves a binary distinction between genes that only have H3K27me3 (accessible to REF6) and genes that also have H3K9me2 and CHGm (inaccessible to REF6). However, the data do not support such a strong conclusion. For example, Figure 2c shows significant but modest differences between control genes (about 90% are not upregulated) and genes with REF6 target domains that lack CHGm (about 80% not upregulated). Furthermore, because CHGm is not absolute and rarely reaches 50%, the effects of CHGm on REF6 recruitment in endosperm are likely to be quantitative rather than qualitative. These limitations should be clearly explained.

7. Statistical approaches are not always deployed appropriately. Throughout the manuscript, statistical comparisons are performed between multiple test groups and a single control group. This is appropriate when differences with the control group are key, but not appropriate when differences between the test groups are of primary interest. In such cases, the authors should use ANOVA or a similar approach that can simultaneously evaluate multiple groups.

An example of this issue is in Figure 2c. The lower four bars indicate that genes without CHGm are significantly upregulated, but not genes with CHGm. However, there are many fewer CHGm genes. The key issue is whether genes with and without CHGm behave differently from each other. A similar example is in Figure 5b. Genes with and without CHGm are likely not different from each other, but genes with CHGm do not reach significance in comparison to the control group because they are fewer (116) than genes without CHGm (488). Figure 8a also suffers from this problem. The top three blue boxes are marked as significant, and the following three grey boxes are not, but the blue and grey boxes are probably not significantly different from each other. The authors should carefully reassess the statistical comparisons and adjust their conclusions accordingly.

8. The regulation of genomic imprinting during germination by REF6 and SUVH4/5/6 is supported by a small number of genes (Figure 8). Similarly, the silencing of paternal alleles of maternally-expressed genes is linked to a very small level of CHH methylation on genes. The authors should acknowledge that regulation of genomic imprinting during germination by REF6 and SUVH4/5/6 is not fully demonstrated by the data shown, and the authors should moderate their claims accordingly.

*Reviewer #2:*

The study by Sato et al. examines the role of chromatin remodelling in the endosperm during dormancy and germination control. The challenge with any study if this kind is that individual epigenetic processes are very general, required at multiple stages for the regulation of seed development, germination and associated metabolism. When a phenotype is observed in mutants lacking a chromatin remodelling process this may be a product of multiple effects, or specific effects on a limited number of loci at one developmental stage. In this study the authors show that the REF6/H3K27me3 suvh456/H3K9me2 interaction is important for gene regulation in the endosperm during germination. I find this interesting and the authors understandably would like to expand this to show that functionally REF6 is required for germination responses in seeds. In my view this aspect of the study is less successful, perhaps reflecting the fact that the authors are world leaders in the field of plant epigenetics but less experienced in the study of seed physiology.

1. Lets start with reviewing what I believe is shown to the highest degree of confidence. In my view the data strongly support the conclusion that the ref6 mutants have germination phenotype which is conferred through endosperm-specific REF6 activity. Although the phenotypes are very weak, the authors take care to generate consistent data across experiments. Cmt3 and suvh456 mutants seem to have a maternal effect on dormancy consistent with previous studies of the maternal roles of these genes in the endosperm. The interaction between H3K27me3, H3K9me2, REF6 and CHG methylation is clearly described on the context of chromatin remodelling during germination, and I agree looks similar in seeds to that shown in a recent study (Qi et al., 2019). It seems clear that REF6-mediated epigenetic remodelling plays a role in the induction of ethylene responsive genes but also for genes in cell wall remodelling, lipid metabolism and other processes known to occur in the endosperm during germination, perhaps in response to ethylene. I would contrast the function of REF6 with that of PICKLE (Ogas et al., 1998) which is required for silencing of the seed expressed genes upon the transition to vegetative growth.

2. There is one experimental issue that make some of the data hard to interpret. The authors would ideally take more care to distinguish clearly between dormancy and germination of non-dormant seeds. Firstly, the dormancy assays are confounded by the fact that a nitrate source was added to the germination medium, a strong dormancy-breaking signal. As such all seed lots behave as if non-dormant in all the experiments, although a difference in germination speed between mutant and wt is recorded. It is not clear from the data provided whether without nitrate the seeds would actually show dormancy, but my feeling is that they would, but only then can it be studied. Certainly, where seeds lots germinate 100% after a few days of imbibition the field would not ascribe any dormancy to them, and would not usually use such experiments to make judgements about dormancy or use the word dormancy to describe the phenotypes. However, the authors rightly intuit that differences in germination speed can result from residual activity of dormancy-inducing pathways even if they have insufficient activity to completely prevent germination (Bassel and Finch-Savage, 2016). Evidence that this is happening is that germination speed is further increased by cold stratification, another dormancy-breaking signal. However, sowing freshly harvested seed on minimal media would result in higher quality data and be the normal approach in the field, showing that seed lots indeed show dormancy (germination <50% is better). Then the authors would clearly be able to show that dormancy rather than germination is affected.

3. The main implied narrative in the abstract that these marks cooperate to regulate dormancy loss through reactivation of maternal copies of ethylene response genes is only partly supported by the data in my view. There is no evidence presented to show that only maternal copies are activated, or describing how imprinting affects the activation rate. So the wider significance was less clear, in contrast to Piskurewicz 2016 who showed that dormancy status affected the degree of imprinting, and that imprinting of these genes affected the rate of protein catabolism or dormancy loss.

4. In my view more supporting evidence is required to show that REF6 directly affects endosperm responses to ethylene. I would inhibit germination with paclobutrazol to remove secondary effects and apply ethylene or a precursor, comparing the transcriptional response of wt and ref6 seeds.

5. If REF6 does affect the ethylene response, more evidence is required to show that this is linked to dormancy loss rather than germination promotion. The latter will occur in both dormant and non dormant seeds, and the seeds examined here are either non-dormant or have unknown dormancy because of the problems mentioned previously. The data in figure 1 only show an increase in the expression of H3K27me3-marked ethylene-inducible genes after testa rupture, ie after the germination process has already begun. This also suggests that the process is linked to the mechanics of germination rather than dormancy. This ought to be manifest in germination speed phenotypes even in non-dormant seeds, eg in a ref6 aba2 double mutant. Conversely, if REF6 is involved in dormancy loss maybe there is no dormancy phenotype until a dormancy-breaking signal is applied.

6. Timing of REF6 activity. Primary dormancy is established during seed development and seed maturation. Thus if seeds have different levels of primary dormancy, with some exceptions the cause is usually found in the processes that occur prior to desiccation. It is unclear when the EPR1:REF6 construct is expressed, whether it is also active prior to desiccation when it may affect the establishment of dormancy, rather than dormancy loss during germination. The unfortunate recent publication (Chen et al., 2020) which shows that REF6 affects ABA metabolism during seed maturation also seems to me to be a plausible mechanism for the ref6 dormancy phenotype. However, disagreement over the cause of the phenotype does not necessarily detract from the fact that REF6 is clearly affecting gene expression in the endosperm during germination, it just means that the significance of the process is less clear for the observed phenotype.

7. Also influencing my conclusions is the observation that the suvh456 effect is not limited to the endosperm but also observable in the embryo, where ABI3 is also higher expressed than wt (Figure 7d). This suggests that suvh456 mutant seeds have higher ABA levels or some global effect on dormancy that affects the whole seed and is capable of up-regulating ABI3 expression. I found the authors' discussion of ABI3 hard to follow. It is surprising that ABI3 reaches maximal expression in the mature seed, but is marked with the repressive H3K27me3. If I understand correctly the authors suggest that in suvh456 ABI3 is further activated by REF6. Again this is surprising because their seeds are germinating and during germination ABI3 expression falls rapidly (if the seeds were dormant then this would be more plausible). So it is possible that ABI3 regulation is several steps removed from the activity of the epigenetic process, or that REF6 acts on ABI3 earlier in seed development. Recent paper Chen et al., 2020 seems to shed some light on this, showing that ref6 seeds have higher ABA content. This would lead to an increase in dormancy and higher ABI3 expression, even in the embryo, because endospermic ABA is transported to the embryo to maintain dormancy (Kang et al., 2015).

So in conclusion I think the authors have discovered an interesting process which could well be very important for gene regulation during germination in the endosperm, but have not convinced me that the observed phenotype is due to disruption of this process. However, there may be other consequences that are observable if endosperm responses to ethylene are investigated in more detail, rather than focussing on the effect on primary dormancy. Alternatively, if the authors wish to explain the germination phenotype it may be worth expanding the study to events during the establishment of primary dormancy, taking into account the recent published observations of Chen et al., 2020.

[Editors' note: further revisions were suggested prior to acceptance, as described below.]

Thank you for resubmitting your work entitled "Combinations of maternal-specific repressive epigenetic marks in the endosperm control seed dormancy" for further consideration by *eLife*. Your revised article has been evaluated by Jürgen Kleine-Vehn (Senior Editor), Daniel Zilberman (Reviewing Editor) and the original reviewers.

The manuscript has been greatly improved and the reviewers and editors are generally satisfied that the revisions address the majors points raised after initial review. A single important remaining issue remains, as explained in the three related major points below. Please consider these points and address them in the manuscript as appropriate.

1. Although they germinate slower than wild type, ref6 mutant response to ACC is similar to wt in terms of the type of germination increase promoted. Certainly ref6 does not appear to have low sensitivity to ACC. Therefore the authors' data suggest that ethylene responses are normal in ref6 mutants. The authors conclude that because ref6 germination deficiency is reversed by ACC (with fluridone), ref6 must be acting through ethylene signalling. How the authors reached this conclusion is not clear.

2. The authors may not be correctly interpretating the aba2 rescue of ref6 in Chen 2020. Chen 2020 show that aba2 more-or-less fully rescues the ref6 phenotype, and indeed any other finding would be very surprising. There are small differences but these are well within normal biological variation. The issue with the fluridone experiment is that fluridone inhibits ABA synthesis in imbibed seeds, and cyp707a2 is important for ABA catabolism in imbibed seeds. But according to Chen et al. REF6 mainly affects CYP707A1 and A3, which are active primarily during seed maturation, and therefore affect germination through the amount of ABA left over in mature seeds (Okamoto et al., 2006). In that context the failure of fluridone to completely suppress the ref6 phenotype, if caused by up-regulation of CYP707A1/3, is unsurprising.

3. New genetic data is presented that shows that the activity of REF6 in the central cell is sufficient to lead to most of the phenotypic effects. The H3K27me3 data used for figure 1 is from developing endosperm, after REF6 could have acted in the gametophyte. But the authors argue that REF6 is affecting de-methylation during dormancy loss or germination, or at least after the stage tissue was collected for the 2016 paper. So is the activity of REF6 described in the rest of the paper the weak phenotype attributed to the paternal effect? Could the differences (which are small) be simply due to gene dosage effects in the endosperm, if REF6 is a rate limiting factor, even if REF6 is not imprinted? If dormancy is determined by central cell REF6 activity, then it can't relate to the effects on imprinted gene expression during germination.

If the above is true, the statement in the manuscript title is mis-leading, because dormancy is determined by central cell ref6 function – the events during germination are clearly affected by REF6, but lack an obvious phenotypic outcome (in Col-0 at least). This is also pertinent to the normal ACC response in ref6 mutants. A clear statement on this issue is hence necessary.

---

## [Author Response]

Essential revisions:1. The reviewers and editors felt that the most important revisions concern the narrative that REF6 and SUVH4-6 control dormancy via ethylene responses. The mechanism by which REF6 regulates dormancy is an essential component of the manuscript. However, the presented mechanism is not convincingly supported by the supplied data and differs from the conclusions of a recently published paper (Chen et al., Plant Physiology 2020), which implicates ABA catabolism during seed development. The authors should establish the relationship between REF6, ethylene and ABA signalling, and ideally reconcile their findings with those of Chen et al., 2020.

To address this concern, we performed several experiments and in silico analyses. First, to test the contribution of the ethylene and ABA pathways to the delayed germination phenotypes in *ref6*, we analyzed germination of *ref6* upon treatment with the ethylene precursor 1-aminocyclopropane-1carboxylic acid (ACC), and the ABA biosynthesis inhibitor fluridone, which were reported to induce seed germination (Linkies et al., 2009; Martinez-Andujar et al., 2011). Treatment with only ACC or fluridone failed to completely suppress the delayed germination of *ref6* (Figure 5—figure supplement 4), consistent with previous data showing that loss of the ABA biosynthetic gene *ABA2* does not completely rescue the delayed germination phenotype of *ref6* after stratification (Chen et al., 2020). In contrast, fluridone could completely suppress the *cyp707a2* germination phenotype (Figure 5—figure supplement 4), revealing that fluridone can effectively suppress ABA biosynthesis as previously shown (Martinez-Andujar et al., 2011). Supporting our transcriptome data showing that ethylene pathways were suppressed in *ref6* (Figure 5c), we found that treatment with both, ACC and fluridone, could completely rescue the delayed germination phenotype of *ref6* after stratification(Figure 5— figure supplement 4). We thus conclude that suppression of the ethylene pathway together with the previously revealed induction of the ABA pathway (Chen et al., 2020) account for the delay of germination in *ref6*. We included and discussed this new data on P11, L243 to L261.

Second, we have further investigated the contribution of SUVH4/5/6 in regulating germination. As shown in our previous version of the manuscript, the *ABI3* gene has triple repressive marks on the maternal alleles and was upregulated in *suvh456* endosperm during germination (Figure 7). We furthermore found that genes with single H3K27me3 tend to be suppressed in *suvh456* endosperm probably through indirect effects (Figure 7—figure supplement 2). In the revised manuscript we include data showing that B3 type transcription factors-binding motifs (RY motifs) are enriched among the downregulated genes with single H3K27me3 in *suvh456* (Figure 7—figure supplement 5). This data suggests that upregulated ABI3 in *suvh456* suppresses genes with single H3K27me3. Supporting this idea, we found that genes with single H3K27me3 are significantly enriched among direct ABI3 target genes (Tian et al., 2020), while genes with single H3K9me2 and double H3K27me3/H3K9me2 are not enriched (Figure 7—figure supplement 5). Together, this data reveals a complex network between ABA and ethylene pathways in the endosperm that are both epigenetically and transcriptionally regulated by multiple factors. We have included this new data on P14, L332 to L338 and discussed on P17, L417 to L418, P18, L431 to L437, P19, L461 to L476 and P21, L525 to L526.

2. Related to the above point, seed dormancy should be assessed in the absence of dormancy-breaking stimuli, especially nitrate in the MS medium. Sowing freshly harvested seed on minimal media would result in higher quality data and be the normal approach in the field, showing that seed lots indeed show dormancy (germination <50% is better). Then the authors would clearly be able to show that dormancy rather than germination is affected.

To address this comment, we performed analyses of seed dormancy on minimal media (Figure 3figure supplement 3a, Figure 6—figure supplement 1a and Figure 6—figure supplement 2).

In addition to that, we further confirmed seed dormancy phenotypes in *ref6* single and *suvh456* triple mutant seeds that were developed under low temperature (Figure 3—figure supplement 3b, Figure 6figure supplement 1c). We included this new data on P8, L177 and P12, L276.

3. The authors are not always clear on the mutant allele used in different experiments. A major caveat is that the ref6-1 and ref6-3 mutants used in germination experiments (Figure 3, Figure 3¬—figure supplement 2 and Figure 3—figure supplement 3) and most of the molecular analyses are not a full knockout (null) mutants (Yan et al., 2019 and Antunez et al., 2020). The use of partial-loss-of-function REF6 alleles could be problematic, in particular for germination assays with seeds from reciprocal crosses. To address this issue, authors should use a loss-of-function mutant line ref6C, which they have used to generate the triple elf6-3 ref6C jmj13G mutant.Similarly, the interpretations arising from the endosperm transcriptome analysis with ref6-1 mutant plants should be carefully considered in their conclusions.

To address this concern, we performed several new experiments and in silico analyses that support our conclusions reached with *ref6-1* and *ref6-3* alleles (see figures below).

In *ref6-1/ref6-3* alleles, the T-DNAs are inserted into the coding region between the JumonjiC domain and Zinc-finger domain, and weak expression of *REF6* mRNA was detected (Yan et al., 2019). This raises the possibility that the Zinc-finger domain-independent functions of REF6 still remain active in *ref6-1* or *ref6-3*. It was previously shown that REF6 can be recruited by MADS-box transcription factors independently of the REF6 binding domain (Yan et al., 2019).

1. We obtained the *ref6c* null allele and compared the phenotype with *ref6-1* and *ref6-3* alleles. (Additional figure 1a), and also analyzed the parental-specific effects of *ref6c* (Additional figure 1b). The results show that the phenotypes of the mutant alleles are highly comparable, justifying the use of the *ref6-1* and *ref6-3* alleles for this study.

2. We investigated the possibility that REF6 is recruited to MADS-box motifs by analyzing whether predicted REF6 targets have REF6 binding sites. As shown in the manuscript, the presence of REF6binding sites is associated with the activation of genes with single H3K27me3 and double H3K27me3/H3K9me2 (Figure 2c). Consistently, REF6-binding sites are significantly enriched within gene bodies of upregulated genes with single H3K27me3 and double H3K27me3/H3K9me2 (additional figure 1c, left panel), while MADS-box binding sites on promoter regions are not enriched among the upregulated genes (additional figure 1c, right panel). Furthermore, we found that there were substantially fewer MADS-box binding sites (around 10%) than REF6-binding sites (40-60%) among upregulated genes. This data suggests that the possible recruitment of REF6 by MADS-box transcription factors in *ref6-1* and *ref6-3* is unlikely to have a large contribution to total gene activation. This data are in support with our observations that the *ref6-1/ref6-3* alleles behave very similar to the *ref6c* allele and justify their use in this study.

4. The interpretation that suvh4/5/6 cause a maternal effect in germination due to the activity of these histone methylases in the central cell is not fully supported. Genetic analyses would be required to distinguish sporophytic and gametophytic effects of these mutants on seed germination. Similarly, the parent-of-origin effects on germination arising in the cmt3 mutant are far from clear. The role of suvh4/5/6 and cmt3 in parent-of-origin effects on germination should be critically assessed and described in the manuscript accordingly.

To address this concern, we analyzed the gametophytic effect of *suvh456* on germination using F1 seeds generated by pollinating heterozygous *suvh456* mutants with wild-type pollen. The results support the conclusion that *suvh456* caused delayed germination through gametophytic effects (Figure 6b). We described this new data on P12, L287 to L290.

Additionally, we analyzed the cause of the maternal effect of *ref6* by analyzing the phenotypes of F1 seeds generated by pollinating heterozygous *ref6-1* with wild-type pollen. The results support the conclusion that the maternal effect of *ref6* is caused by a gametophytic effect (Figure 3c). We described this new data on P8, L183 to P9, L191.

The *cmt3-11* mutant did not have a maternal effect on germination (Figure 6d), indicating that CMT3 acts after fertilization on H3K9me2 marks established in the female gametophyte. One possible explanation for the difference between *suvh456* and *cmt3* mutants is different effect on H3K9me2; while loss of *SUVH4/5/6* causes severe depletion of both, H3K9me2 and CHGm (Stroud et al., 2014; Underwood et al., 2018), loss of *CMT3* causes depletion of CHGm but has only intermediate effects on on H3K9me2 (Stroud et al., 2014; Inagaki et al., 2010). This suggests that genes with H3K27me3/H3K9me2 are suppressed by the paternal CMT3 after fertilization. We have discussed this data on P13, L298 to L301 and P20, L485 to L489.

5. The expression of REF6 in the micropylar endosperm and embryo using EPR1::REF6 and TPS1::REF6 should be confirmed experimentally (eg. RT-PCR with dissected tissues).

We confirmed the tissue-specific expression of *REF6* in *EPR1::REF6;ref6-1* and *TPS1::REF6;ref6-1* (Figure 4—figure supplement 1).

6. The strong claims about the effects of CHGm on REF6 should be moderated. The model communicated by the authors involves a binary distinction between genes that only have H3K27me3 (accessible to REF6) and genes that also have H3K9me2 and CHGm (inaccessible to REF6). However, the data do not support such a strong conclusion. For example, Figure 2c shows significant but modest differences between control genes (about 90% are not upregulated) and genes with REF6 target domains that lack CHGm (about 80% not upregulated). Furthermore, because CHGm is not absolute and rarely reaches 50%, the effects of CHGm on REF6 recruitment in endosperm are likely to be quantitative rather than qualitative. These limitations should be clearly explained.

To address this concern, we moderated our claim about the relationship between CHGm directly on REF6 binding sites and REF6 accessibility. As the editor suggests about 10% of genes with H3K27me3/H3K9me2 and CHGm on REF6 binding sites are induced during germination, suggesting unknown mechanisms can remove the triple repressive marks or paternal alleles of some of those genes are activated during germination. It will be interesting to study whether and how the strong suppressing mechanism can be released. More detailed analyses of individual genes with triple repressive marks will be required to reveal the complete molecular mechanisms establishing and removing triple repressive marks. We have discussed these points on P17, L418 to L420.

7. Statistical approaches are not always deployed appropriately. Throughout the manuscript, statistical comparisons are performed between multiple test groups and a single control group. This is appropriate when differences with the control group are key, but not appropriate when differences between the test groups are of primary interest. In such cases, the authors should use ANOVA or a similar approach that can simultaneously evaluate multiple groups.An example of this issue is in Figure 2c. The lower four bars indicate that genes without CHGm are significantly upregulated, but not genes with CHGm. However, there are many fewer CHGm genes. The key issue is whether genes with and without CHGm behave differently from each other. A similar example is in Figure 5b. Genes with and without CHGm are likely not different from each other, but genes with CHGm do not reach significance in comparison to the control group because they are fewer (116) than genes without CHGm (488). Figure 8a also suffers from this problem. The top three blue boxes are marked as significant, and the following three grey boxes are not, but the blue and grey boxes are probably not significantly different from each other. The authors should carefully reassess the statistical comparisons and adjust their conclusions accordingly.

To address this concern, we reanalyzed our data using pairwise statistical tests with multiple testing correction. The statistical methods are described in the legend of each figure.

8. The regulation of genomic imprinting during germination by REF6 and SUVH4/5/6 is supported by a small number of genes (Figure 8). Similarly, the silencing of paternal alleles of maternally-expressed genes is linked to a very small level of CHH methylation on genes. The authors should acknowledge that regulation of genomic imprinting during germination by REF6 and SUVH4/5/6 is not fully demonstrated by the data shown, and the authors should moderate their claims accordingly.

To address this concern, we moderated our claims and specifically noted that further studies are required to validate the conclusions. Modifications are on P16, L376 to L377, L395 to L397 and P18, L441 to L443.

We also reanalyzed the DNA methylation data using pairwise Wilcoxon test and included up-and downstream regions in the analyses (Figure 8—figure supplement 2, 3 and 4). Also for this analyses we toned down the conclusions and included the modifications on P16, L383 to L397.

[Editors' note: further revisions were suggested prior to acceptance, as described below.]

Essential revisions:1. Although they germinate slower than wild type, ref6 mutant response to ACC is similar to wt in terms of the type of germination increase promoted. Certainly ref6 does not appear to have low sensitivity to ACC. Therefore the authors' data suggest that ethylene responses are normal in ref6 mutants. The authors conclude that because ref6 germination deficiency is reversed by ACC (with fluridone), ref6 must be acting through ethylene signalling. How the authors reached this conclusion is not clear.

We agree with the reviewer that the *ref6* mutant can respond to ethylene; however, also ethylene biosynthesis is negatively affected in ref6 mutants (Figure 5—figure supplement 3); thus by adding ACC, ethylene production likely increases and thus ethylene signalling. Furthermore, since ethylene biosynthesis and signalling are in a feedback loop and ethylene was shown to directly regulate its biosynthesis (doi.org/10.1111/nph.16873), it is well possible that by adding ACC and thus highly increasing ethylene levels, ethylene signalling is enhanced. Since the combined treatment of fluridone and ACC completely suppressed the *ref6* mutant phenotype, we consider our interpretation that REF6 acts (as well) through ethylene signalling justified. We discussed the requirement of REF6 to induce ethylene biosynthesis on P11, L239 to P11, L244.

2. The authors may not be correctly interpretating the aba2 rescue of ref6 in Chen 2020. Chen 2020 show that aba2 more-or-less fully rescues the ref6 phenotype, and indeed any other finding would be very surprising. There are small differences but these are well within normal biological variation. The issue with the fluridone experiment is that fluridone inhibits ABA synthesis in imbibed seeds, and cyp707a2 is important for ABA catabolism in imbibed seeds. But according to Chen et al. REF6 mainly affects CYP707A1 and A3, which are active primarily during seed maturation, and therefore affect germination through the amount of ABA left over in mature seeds (Okamoto et al., 2006). In that context the failure of fluridone to completely suppress the ref6 phenotype, if caused by up-regulation of CYP707A1/3, is unsurprising.

The data from Chen et al., 2000 show that the *ref6 aba2* double mutant has lower germination rates than the *aba2* single mutant after stratification (Figure 2C). While the difference has not been statistically evaluated, a germination difference of 50% versus 90% at 1.5 DAP seems to be highly significant. Since the data are generated in biological triplicates; we consider it unlikely that the difference is solely due to biological variation.

3. New genetic data is presented that shows that the activity of REF6 in the central cell is sufficient to lead to most of the phenotypic effects. The H3K27me3 data used for figure 1 is from developing endosperm, after REF6 could have acted in the gametophyte. But the authors argue that REF6 is affecting de-methylation during dormancy loss or germination, or at least after the stage tissue was collected for the 2016 paper. So is the activity of REF6 described in the rest of the paper the weak phenotype attributed to the paternal effect? Could the differences (which are small) be simply due to gene dosage effects in the endosperm, if REF6 is a rate limiting factor, even if REF6 is not imprinted? If dormancy is determined by central cell REF6 activity, then it can't relate to the effects on imprinted gene expression during germination.If the above is true, the statement in the manuscript title is mis-leading, because dormancy is determined by central cell ref6 function – the events during germination are clearly affected by REF6, but lack an obvious phenotypic outcome (in Col-0 at least). This is also pertinent to the normal ACC response in ref6 mutants. A clear statement on this issue is hence necessary.

While *ref6* has a maternal effect, the fact that many REF6 targets have detectable levels of H3K27me3 in the endosperm and since we could partially complement the *ref6* mutant by expressing *REF6* in the micropylar domain of the endosperm (Figure 5a), strongly suggests that REF6 also acts after fertilization. In support of this, homozygous *ref6* mutants have a stronger delay in germination than mutants inheriting a maternal *ref6* and a wild-type paternal allele (Figure 3b). Nevertheless, independent of when REF6 acts; the consequence of REF6 activity occurs during germination. We had discussed the idea that there are two types of genes that are either activated in the central cell or later on in the endosperm in the manuscript lines 472 to 495. In summary, we do not think that our title is misleading, the combination of epigenetic modifications present in the endosperm (independent on when they are established) determines the activity status of the alleles during germination.